# TOWARDS A FOUNDATION MODEL FOR CROWD-SOURCED LABEL AGGREGATION

**Hao Liu[1], Jiacheng Liu[2,†], Feilong Tang[3,†], Long Chen[4], Jiadi Yu[3], Yanmin Zhu[3],**
**Qiwen Dong[1,†], Yichuan Yu[5], Xiaofeng Hou[3]**
[1]East China Normal University
[2]The Hong Kong University of Science and Technology
[3]Shanghai Jiao Tong University
[4]Simon Fraser University
[5]Huawei Technologies

## ABSTRACT

Inferring ground truth from noisy, crowdsourced labels is a fundamental challenge in machine learning. For decades, the dominant paradigm has relied on dataset-specific parameter estimation, a non-scalable method that fails to transfer knowledge. Recent efforts toward universal aggregation models do not account for the structural and behavioral complexities of human-annotated crowdsourcing, resulting in poor real-world performance. To address this gap, we introduce CrowdFM, a foundation model for crowdsourced label aggregation. At its core, CrowdFM is a bipartite graph neural network that is pre-trained on a vast, domain-randomized synthetic dataset to learn diverse behavioral patterns. By leveraging a size-invariant initialization and attention-based message passing, it learns universal principles of collective intelligence and generalizes to new, unseen datasets. Extensive experiments on 22 real-world benchmarks show that our single, fixed model consistently matches or surpasses bespoke, per-dataset methods in both accuracy and efficiency. Furthermore, the representations learned by CrowdFM readily support diverse downstream applications, such as worker assessment and task assignment. Codes are available at https://github.com/liiuhaao/CrowdFM.

## 1 INTRODUCTION

Crowdsourcing enables efficient collection of large-scale labeled datasets by distributing annotation tasks to non-expert workers (Howe et al., 2006). However, variations in worker expertise often produce noisy and conflicting labels (Sheng & Zhang, 2019). Label aggregation serves as a fundamental step to infer reliable ground truth from such inputs (Zheng et al., 2017; Zhang et al., 2024b;a).

For decades, two conflicting paradigms have dominated this field. On one hand, Majority Voting (MV) stands as the de facto industry standard for scalable deployment (Penrose, 1946). Its unmatched simplicity ensures high scalability and, crucially, a *retraining-free nature*, allowing it to be applied universally across any dataset. However, its assumption of uniform worker quality leads to suboptimal accuracy in heterogeneous environments (Tullock, 1959). On the other hand, in pursuit of higher accuracy, a vast body of advanced methods has emerged, from probabilistic models (Dawid & Skene, 1979; Whitehill et al., 2009) to deep learning approaches (Li'ang Yin et al., 2017; Wu et al., 2023a; Liu et al., 2024). Yet, these methods sacrifice the very properties that make MV practical. They are all confined to a *dataset-specific paradigm*: they require learning dataset-specific parameters from scratch, rendering them non-scalable, brittle, and unable to transfer.

Bridging this gap requires a model that combines the accuracy of advanced methods with the scalability and universality of MV. A promising solution is a foundation model for crowdsourced label aggregation that generalizes across datasets without dataset-specific retraining. HyperLM (Wu et al., 2023b) pioneered cross-dataset GNN aggregation but fails to address the structural heterogeneity in-

---

†Corresponding authors.

trinsic to crowdsourcing settings. Specifically, its architecture lacks explicit worker-task modeling and its training relies on uniform synthetic data that misaligns with real-world patterns.

To overcome these limitations, realizing a truly transferable crowdsourcing foundation model faces two main challenges. First, *universal crowdsourcing representation*: datasets vary widely in the number of workers, tasks, and label options, and may exhibit diverse annotation patterns. A universal model must flexibly encode any such configuration into a meaningful representation that captures worker and task heterogeneity, enabling effective generalization to downstream adaptation. Second, *realistic synthetic data generation*: foundation models require large-scale pre-training resources, but existing crowdsourcing datasets are limited. The training data must faithfully reflect real-world crowdsourcing patterns to support robust transfer.

In response to these challenges, we introduce CrowdFM, a foundation model for crowdsourced label aggregation. CrowdFM employs a graph neural network that explicitly models workers, tasks, and label options, naturally capturing structural heterogeneity across crowdsourcing datasets. To address realistic synthetic data generation, we design a domain-randomized generator that creates diverse scenarios closely matching real crowdsourcing datasets. Pre-training on this data enables CrowdFM to learn robust, transferable aggregation principles. The main contributions of this work are summarized as follows:

- We propose CrowdFM, a foundation model based on a bipartite graph neural network for crowdsourced label aggregation that explicitly represents workers, tasks, and options, achieving generalization across heterogeneous datasets without dataset-specific training.

- We develop a synthetic crowdsourcing data generator that produces diverse datasets reflecting real-world crowdsourcing patterns, enabling the model to pretrain on varied scenarios and learn aggregation principles transferable to unseen datasets.

- Extensive experiments show that CrowdFM achieves performance comparable to or superior to state-of-the-art aggregation methods, while remaining efficient, stable, and retraining-free. In addition, it can be readily applied to multiple downstream crowdsourcing applications, highlighting its versatility in practical workflows.

## 2 PRELIMINARY

**Crowdsourced data.** We consider a crowdsourcing setting with $N$ tasks $\mathcal{T} = \{t_1, \ldots, t_N\}$ and $M$ workers $\mathcal{W} = \{w_1, \ldots, w_M\}$, where each task $t_j$ has a true label $y_j \in \mathcal{O} = \{1, \ldots, K\}$. Each annotation $a_{ij}$ is an observed label provided by worker $w_i$ on task $t_j$, forming the annotation set $\mathcal{A} = \{(w_i, t_j, a_{ij})\}$. The goal of label aggregation is to infer the true labels $\mathcal{Y} = \{y_j\}_{j=1}^N$ from $\mathcal{A}$.

**Dataset-Specific modeling in conventional approach.** Conventional approaches adopt a per-dataset modeling paradigm: for each dataset $\mathcal{D}^{(s)} = (\mathcal{A}^{(s)}, \mathcal{Y}^{(s)})$, they define a generative model $p(\mathcal{A}^{(s)} \mid \Theta^{(s)})$ and estimate dataset-specific latent variables $\Theta^{(s)}$ via maximum likelihood:

$$\Theta^{(s)*} = \arg\max_{\Theta^{(s)}} \log p(\mathcal{A}^{(s)} \mid \Theta^{(s)}), \quad \hat{\mathcal{Y}}^{(s)} = \mathcal{I}^{(s)}(\mathcal{A}^{(s)}; \Theta^{(s)*}), \tag{1}$$

where $\Theta^{(s)*}$ denotes the learned latent variables such as worker abilities and task difficulties for dataset $D^{(s)}$, and $\mathcal{I}^{(s)}$ represents the inference procedure applied within the same dataset using the fitted model. Since no parameters are shared across datasets, these models must be re-estimated from scratch for every new deployment.

**Cross-dataset generalization in our Approach.** In contrast, we follow a cross-dataset generalization paradigm. We learn a single, parameter-shared aggregation function $F_{\Theta} : \mathcal{A} \mapsto \hat{\mathcal{Y}}$ by minimizing the expected loss over a distribution of potential datasets $p_{\mathcal{D}}$:

$$\Theta^* = \arg\min_{\Theta} \mathbb{E}_{\mathcal{D}(\mathcal{A}, \mathcal{Y}) \sim p_{\mathcal{D}}} \left[ \ell(F_{\Theta}(\mathcal{A}), \mathcal{Y}) \right], \quad \hat{\mathcal{Y}}' = F_{\Theta^*}(\mathcal{A}'). \tag{2}$$

The model $F_{\Theta^*}$ is fixed after pretraining and deployed zero-shot on new datasets without re-estimation. This design enables direct, retraining-free inference across diverse domains, fundamentally differing from per-dataset estimation.

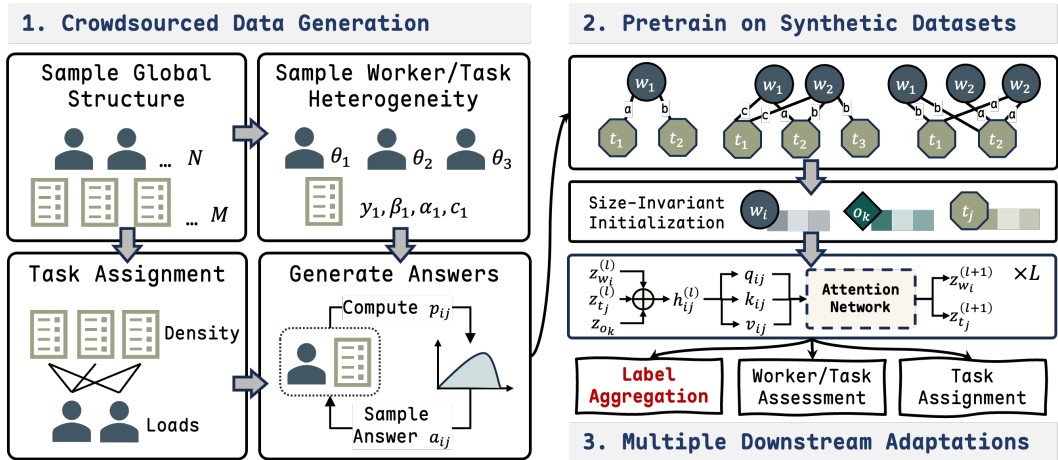

Figure 1: The overall framework of CrowdFM.

# 3 METHODOLOGY

We propose CrowdFM, a framework for learning generalizable crowdsourcing aggregation models through synthetic data pretraining. The overall pipeline, illustrated in Figure 1, comprises three key components: generating diverse synthetic crowdsourced datasets, pretraining the model on these datasets to learn robust aggregation rules, and applying the pretrained model for label aggregation as well as other downstream applications.

## 3.1 SYNTHETIC CROWDSOURCED DATA GENERATOR

Public available real-world crowdsourcing datasets are highly scarce, making them insufficient for pretraining robust aggregation models. To address this, we design a synthetic data generator that produces diverse crowdsourcing datasets by randomizing key aspects of the data, including global structure, worker and task characteristics, assignment patterns, and annotation generation.

**Global structural randomization.**    To ensure that models can adapt to crowdsourcing scenarios of varying scales and sparsity levels, we randomize the global structure of each synthetic dataset. This exposes models to environments with diverse combinations of size and labeling density, encouraging the development of robust aggregation behaviors. In practice, for each dataset we independently sample the number of tasks $N$, workers $M$, answer options $K$, and the expected number of annotations per task $A$ from broad, realistic ranges.

**Behavioral heterogeneity.**    Real-world crowdsourcing datasets consistently show significant variation in worker reliability and task difficulty, both of which strongly influence annotation quality. To model this heterogeneity, we sample worker ability as $\theta_i \sim \mathcal{N}(\mu_\theta, \sigma_\theta^2)$. Task difficulty $\beta_j$ is drawn from $\mathcal{N}(\mu_\beta, \sigma_\beta^2)$, task discrimination $\alpha_j$ from $\text{Uniform}(\alpha_{\min}, \alpha_{\max})$, and task guessing rate $c_j$ from $\text{Uniform}(1/K, c_{\text{upper}})$. Each task's true label $y_j$ is sampled from $\{1, \ldots, K\}$. Notably, the parameters governing these distributions are themselves randomly drawn for each dataset, producing heterogeneous worker and task profiles and more realistic annotations.

**Task assignment mechanism.**    Crowdsourcing typically exhibits long-tailed participation, where a few workers contribute many labels while most contribute only a few. To capture this, we draw each worker's labeling capacity $L_i$ from a heavy-tailed distribution, ensuring heterogeneous participation levels across workers. For each task $t_j$, the number of assigned annotators $n_j$ is drawn from a Poisson distribution parameterized by $A$, capturing the variability in annotation density across tasks. Annotators for each task are selected from workers with remaining capacity. This mechanism produces naturally uneven annotation coverage, exposing models to diverse participation patterns that mirror real-world data.

**Annotation generation via the 3PL response model.** Observed annotation labels should reflect both worker skill and task properties, while still allowing for random errors. We adopt the three-parameter logistic (3PL) model from Item Response Theory (DeMars, 2010) to capture this. For each worker–task pair $(w_i, t_j)$, the probability of a correct annotation is

$$p_{ij} = c_j + (1 - c_j) \cdot \sigma(D\alpha_j(\theta_i - \beta_j)), \tag{3}$$

where $\sigma(x) = (1 + e^{-x})^{-1}$ denotes the logistic function and $D$ is a scaling constant. The observed label $a_{ij}$ is generated according to this probability, taking the true label $y_j$ with probability $p_{ij}$, and with probability $1 - p_{ij}$, an incorrect option is randomly chosen from the remaining $K - 1$ labels.

**Generated dataset output.** The final synthetic dataset is $\mathcal{D} = (\mathcal{A}, \mathcal{Y})$, where $\mathcal{A} = \{(w_i, t_j, a_{ij})\}$ contains all collected annotations and $\mathcal{Y} = \{y_j\}$ contains the corresponding ground-truth labels. The full generation procedure is summarized in Algorithm 1. Each dataset instance is independently sampled from broad distributions, exposing models to varying scales, sparsity levels, and noise patterns. This encourages the learning of aggregation rules that generalize across datasets rather than overfitting to a fixed setting.

### 3.2 GNN STRUCTURE FOR CROSS-DATASET GENERALIZATION

**Initialization of representations.** In real-world crowdsourcing datasets, workers and tasks typically have no additional features beyond their IDs. Traditional aggregation methods often rely on dataset-specific information, such as annotation statistics or one-hot identity features, which limits them to the given dataset and prevents generalization to datasets of different sizes. To overcome this, our model adopts a size-invariant initialization strategy: all worker nodes share the same learnable vector $x_w \in \mathbb{R}^d$, all task nodes share another vector $x_t \in \mathbb{R}^d$, and option nodes are independently initialized for each category from a fixed-dimensional Gaussian distribution.

$$z_{w_i}^{(0)} = x_w \ (\forall w_i \in \mathcal{W}), \quad z_{t_j}^{(0)} = x_t \ (\forall t_j \in \mathcal{T}), \quad z_{o_k}^{(0)} \sim \mathcal{N}(0, I_d) \ (\forall o_k \in \mathcal{O}). \tag{4}$$

This design reflects the idea that workers and tasks are indistinguishable before any annotations are observed, and their differences emerge once relational information is incorporated, without introducing dataset-specific priors. Meanwhile, The random initialization of option embeddings ensures sufficient diversity to distinguish among candidate labels regardless of the number of options.

**Attention-based encoder.** Starting from the size-invariant initialization, workers and tasks are treated as nodes in a heterogeneous graph. Rather than learning dataset-specific embeddings, the model allows their representations to evolve solely through relational evidence encoded in observed annotations. The encoder propagates information across the graph with $L$ layers of attention-based aggregation, gradually differentiating otherwise identical worker and task nodes according to their annotation patterns.

At layer $l$, for each observed annotation $(w_i, t_j, a_{ij}) \in \mathcal{A}$, we construct a triple representation:

$$h_{ij}^{(l)} = [\, z_{w_i}^{(l)}, \ z_{t_j}^{(l)}, \ z_{a_{ij}} \,] \in \mathbb{R}^{3d}, \tag{5}$$

and compute queries, keys, and values via type-specific linear projections:

$$q_{ij} = W_q h_{ij}^{(l)} + b_q, \quad k_{ij} = W_k h_{ij}^{(l)} + b_k, \quad v_{ij} = W_v h_{ij}^{(l)} + b_v. \tag{6}$$

Attention weights are obtained by scaled dot-product and normalized over all annotations incident to the same center node:

$$\alpha_{ij}^{(l)} = \mathrm{softmax}\left( \frac{\langle q_{ij}, k_{ij} \rangle}{\sqrt{d}} \right). \tag{7}$$

The embeddings of workers and tasks are then updated by aggregating values with these attention weights, followed by residual connection and layer normalization:

$$z^{(l+1)} = \mathrm{LayerNorm}\left( z^{(l)} + \sum_{(i,j) \in \mathcal{N}} \alpha_{ij}^{(l)} \, v_{ij} \right), \tag{8}$$

where the summation is restricted to the neighborhood $\mathcal{N}$ of the corresponding worker or task node.

Repeating this process for $L$ layers allows the model to encode progressively richer relational patterns, enabling inference over workers and tasks without introducing dataset-specific priors.

**Aggregated label prediction.** After obtaining refined embeddings from the encoder, we compute aggregated label predictions by combining task and option embeddings. For each task $t_j$, its embedding $z_{t_j}$ is concatenated with all option embeddings $z_o$ and passed through a shared feed-forward network to produce logits for each task-option pair:

$$\hat{l}_{jk} = g([\, z_{t_j},\, z_{o_k}\,]). \tag{9}$$

This design allows the model to handle tasks with any number of candidate options, while capturing task-option interactions to facilitate accurate label aggregation. The predicted probability distribution over options for task $t_j$ is then obtained via softmax, and the aggregated label is selected as the option with the highest probability:

$$\hat{p}_{jk} = \frac{\exp(\hat{l}_{jk})}{\sum_{k'=1}^{K}\exp(\hat{l}_{jk'})}, \quad \hat{y}_j = \arg\max_k \hat{p}_{jk}. \tag{10}$$

### 3.3 Pretraining and Inference

**Pretraining on synthetic data.** To enable cross-dataset generalization, the model is pretrained on a collection of annotation datasets $\left\{\mathcal{D}^{(s)} = (\mathcal{A}^{(s)}, \mathcal{Y}^{(s)})\right\}_{s=1}^{S}$ produced by the synthetic data generator (Section 3.1). The encoder and the prediction network are optimized jointly by minimizing the average cross-entropy loss across all synthetic datasets:

$$\mathcal{L} = -\frac{1}{S}\sum_{s=1}^{S}\left(\sum_{j=1}^{N_s}\sum_{k=1}^{K_s}\mathbf{1}[y_j^{(s)} = k]\log\hat{p}_{jk}^{(s)}\right), \tag{11}$$

where $N_s$ and $K_s$ are the number of tasks and options in dataset $\mathcal{D}^{(s)}$, respectively. The function $\mathbf{1}[\cdot]$ denotes the indicator function, which equals 1 if the condition is true and 0 otherwise. The term $\hat{p}_{jk}^{(s)}$ denotes the predicted probability of option $o_k$ for task $t_j$ in dataset $D^{(s)}$.

**Inference on new datasets.** We denote the aggregation pipeline, including embedding initialization, message passing, and prediction, as a function $F(\cdot)$. After pretraining, the parameters are fixed to obtain $F^*(\cdot)$. Given a new annotation set $\mathcal{A}'$, the aggregated labels are computed as:

$$\hat{\mathcal{Y}} = F^*(\mathcal{A}'), \tag{12}$$

where $\hat{\mathcal{Y}} = \{\hat{y}_j\}_{j=1}^{N}$ denotes the set of predicted labels. This procedure enables direct deployment to unseen datasets without any further training or parameter updates.

## 4 Experiments

Our experimental evaluation spans both the main paper and the appendices. Section 4.2 presents label aggregation accuracy results on 22 real-world crowdsourcing datasets. Section 4.3 investigates CrowdFM's ability to learn transferable representations of workers and tasks that generalize to diverse downstream adaptation scenarios. Section 4.4 provides ablation studies and sensitivity analyses of key model components and hyperparameters. Appendix F includes a quantitative analysis comparing synthetic and real-world datasets. Appendix G evaluates CrowdFM's performance under a range of synthetic dataset configurations.

### 4.1 Experimental Setup

Our model is trained on dynamically generated synthetic crowdsourced datasets, where diverse configurations are sampled at each training step to expose the model to a wide variety of scenarios. Implementation details of the synthetic dataset generation, along with the specific parameter ranges

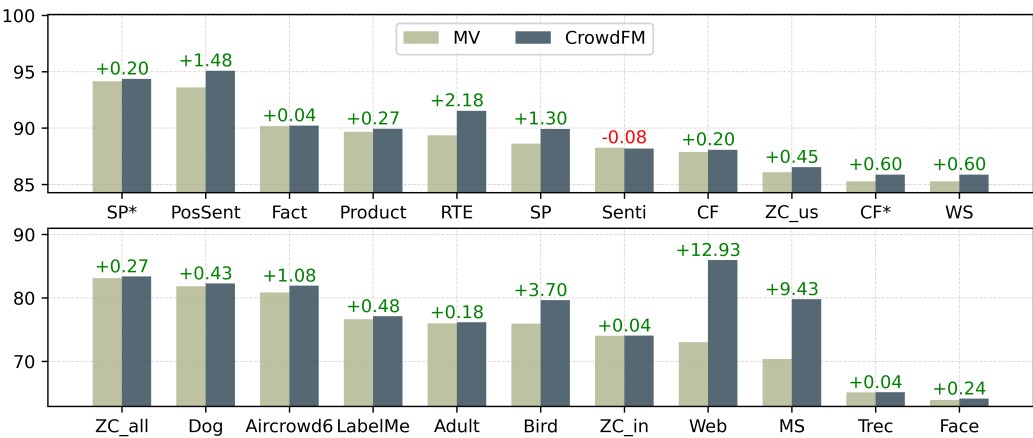

Figure 2: Accuracy comparison between MV and CrowdFM across 22 datasets. The numbers above the bars indicate the performance improvement.

used, are provided in Appendix B. For evaluation, we consider 22 real-world datasets drawn from practical human-labeled scenarios across various domains, with full details of each dataset included in Appendix D. Since all of the considered datasets are intended for classification tasks, accuracy is adopted as the primary evaluation metric, which is defined as the proportion of correctly predicted labels among all instances.

## 4.2 MAIN RESULTS

**Comparison with majority voting.** We conduct a comprehensive evaluation of our framework on 22 real-world crowdsourcing datasets, comparing its performance against the majority voting (MV) baseline. As shown in Figure 2, our method consistently outperforms MV across most datasets, yielding an average improvement of +1.64 percentage points in accuracy. This consistent superiority highlights the strong generalization capability of our approach, capturing the inherent heterogeneity of workers and tasks, while effectively transferring the aggregation strategy learned from synthetic data to real-world scenarios.

Notably, our model delivers particularly large gains on several benchmarks. The most significant improvements are observed on the *Web* and *MS* datasets, where CrowdFM surpasses MV by +12.93% and +9.43%, respectively. Substantial enhancements are also achieved on *Bird* (+3.70%) and *RTE* (+2.18%), indicating the effectiveness of our model in aggregating reliable labels. Across the remaining datasets, CrowdFM maintains consistently strong performance, with only a marginal drop on *Senti* of 0.08%, even though *Senti* deviates from our synthetic training data (Appendix F), highlighting its robustness across diverse crowdsourcing scenarios.

| | #Win↑ | Acc.↑ | Runtime↓ | P-value |
|---|---|---|---|---|
| MV | - | 81.78 | **0.04** | 0.00003 |
| PM | 13 | 80.27 | 0.47 | 0.00647 |
| CATD | 15 | 83.06 | 2.59 | 0.20700 |
| DS | 16 | 83.02 | 5.24 | 0.31889 |
| BWA | 17 | 83.31 | 0.10 | 0.60871 |
| IBCC | 15 | 83.07 | 0.12 | 0.36658 |
| EBCC | 17 | **84.08** | 2.95 | 0.90089 |
| GLAD | 16 | 82.75 | 494.26 | 0.19475 |
| LAA | 10 | 78.42 | 223.06 | 0.04935 |
| TiReMGE | 6 | 80.29 | 26.77 | 0.00230 |
| GOVERN | 13 | 82.61 | 95.43 | 0.28992 |
| HyperLM | 12 | 80.81 | 0.88 | 0.01793 |
| CrowdFM | **21** | 83.41 | 0.53 | - |

Table 1: Performance comparison across 22 real-world crowdsourcing datasets. Win counts indicate the number of datasets where each method outperforms MV. Accuracy and runtime are averaged over all successfully completed runs (LAA and GOVERN failed on several large datasets due to extremely high memory requirements). The P-value reports the one-sided $p$-value from the Wilcoxon signed-ranks test comparing each method against CrowdFM, with lower values indicating stronger significance in difference. Full per-dataset results are provided in Appendix E.

Importantly, several datasets exhibit domain shifts relative to the synthetic training data, such as differences in size and annotation density (see Appendix D). Despite these shifts, CrowdFM maintains performance comparable to or better than MV, consistently demonstrating strong generalization across varying scales and noise patterns. In contrast, many dataset-specific methods often underperform MV (Appendix E.1), as they tend to overfit to particular structures and fail to generalize to other datasets.

**Comparison with dataset-specific methods.** We further compare CrowdFM against a range of state-of-the-art dataset-specific aggregation methods. As shown in Table 1, our method achieves the highest number of wins over MV, outperforming MV on 21 out of 22 benchmarks. This demonstrates its strong competitiveness and broad applicability in diverse crowdsourcing scenarios. Among the dataset-specific methods, BWA and EBCC achieve the next highest number of wins with 17 datasets. GOVERN, the top deep learning-based method, wins on 13 datasets. Despite these performances, none match the consistent superiority of CrowdFM across the full set of datasets.

Specifically, CrowdFM achieves an average accuracy of 83.41%, which is competitive with top-performing models such as EBCC (84.08%) and superior to others including BWA and DS. To assess the statistical significance of these improvements, we conduct the one-sided Wilcoxon signed-ranks test (Demšar, 2006) between CrowdFM and each baseline. The results show that CrowdFM is significantly better than MV, PM, LAA, TiReMGE, and HyperLM. These significant gains confirm the effectiveness of our method. Notably, despite EBCC's marginally higher average accuracy, the performance differences are not statistically significant ($p = 0.90089$), and EBCC incurs a significantly higher computational cost (2.95 seconds per dataset), whereas CrowdFM maintains efficient inference at only 0.53 seconds per dataset, comparable to lightweight methods.

HyperLM, a training-free method designed for programmatic weak supervision, fails to adapt to crowdsourcing settings. It outperforms MV on only 12 out of 22 datasets and achieves an average accuracy of 80.81, which is notably lower than that of our approach (83.41) and even underperforming MV (81.78). With an average runtime of 0.88 seconds, it is notably slower than CrowdFM, which runs in 0.53 seconds. HyperLM also exhibits poor scalability, taking 16.72 seconds on large-scale datasets such as Senti, compared to our method's 5.75 seconds (Appendix E.2). These results confirm that HyperLM is neither as accurate nor as efficient as CrowdFM and is ill-suited for real-world crowdsourcing applications.

Despite being a deep learning-based method, CrowdFM demonstrates favorable runtime efficiency, running much faster than other deep learning approaches such as LAA (223.06 s), TiReMGE (26.77 s), and GOVERN (91.46 s), while remaining comparable in speed to simpler, lightweight methods such as PM (0.47 s). This efficiency and robustness arise from CrowdFM's ability to learn a generalizable aggregation mechanism capturing real-world crowdsourcing patterns during pretraining, eliminating the need for dataset-specific training or iterative parameter estimation at inference time.

## 4.3 MULTIPLE DOWNSTREAM ADAPTATIONS

Having demonstrated strong performance in label aggregation, we now explore the broader utility of CrowdFM beyond its primary task. By design, our model learns to capture the heterogeneity inherent in crowdsourcing environments, learning rich representations of worker behavior and task characteristics through training on diverse synthetic data. This ability aligns with the foundation model paradigm, where a single pretrained network acquires transferable knowledge applicable to multiple downstream adaptations.

For adapting multiple downstream applications, we keep the encoder fixed and only train lightweight downstream-specific heads. These heads are trained once and can be directly deployed on new datasets without further adaptation. This design ensures that CrowdFM remains efficient and deployment-friendly, enabling rapid extension to diverse downstream applications while preserving the pretrained knowledge.

### 4.3.1 WORKER AND TASK ASSESSMENT

In crowdsourcing scenarios, identifying skilled workers and difficult tasks is essential for ensuring annotation quality (Daniel et al., 2018). In this section, we assess the transferability of the learned

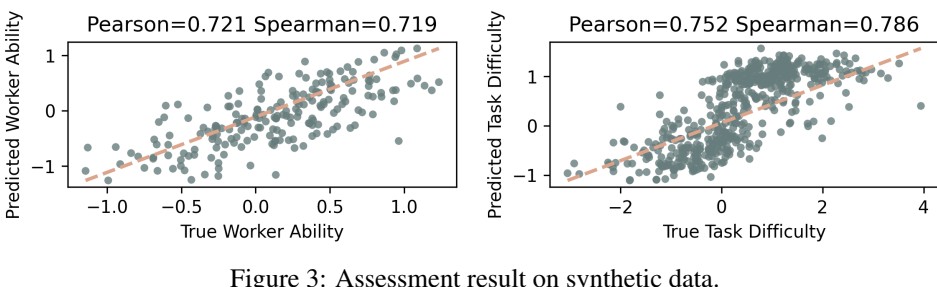

Figure 3: Assessment result on synthetic data.

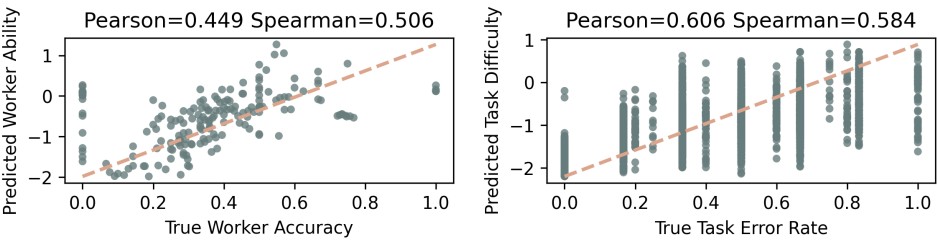

Figure 4: Assessment result on real-world data (Web).

representations from pre-training by applying them to two core quality assessment tasks: estimating global worker ability and inferring global task difficulty.

**Adaptation setup.** We extract worker and task embeddings produced by CrowdFM during inference and use them as features to train lightweight regression heads, supervised by the ground-truth from the synthetic data generator introduced in Section 3.1:

$$\hat{a}_i = g_{\mathrm{a}}(z_{w_i}), \quad \mathcal{L}_{\mathrm{a}} = \frac{1}{|\mathcal{W}|} \sum_{w_i \in \mathcal{W}} (\hat{a}_i - \theta_i)^2, \quad \hat{d}_j = g_{\mathrm{d}}(z_{t_j}), \quad \mathcal{L}_{\mathrm{d}} = \frac{1}{|\mathcal{T}|} \sum_{t_j \in \mathcal{T}} \left( \hat{d}_j - \beta_j \right)^2,$$

(13)

where $\hat{a}_i$ and $\hat{d}_j$ denote the predicted ability of worker $w_i$ and difficulty of task $t_j$, respectively; $z_{w_i}$ and $z_{t_j}$ are the corresponding learned embeddings; $\theta_i$ and $\beta_j$ are the ground-truth values from the synthetic generator. The losses $\mathcal{L}_{\mathrm{a}}$ and $\mathcal{L}_{\mathrm{d}}$ are mean squared errors (MSE) computed over all workers $\mathcal{W}$ and all tasks $\mathcal{T}$, respectively.

**Performance on synthetic data.** We evaluate worker and task assessment performance on synthetic data where ground truth is available. The Pearson and Spearman correlation scores show strong agreement between the predicted and true values (Figure 3), indicating that the model successfully captures latent heterogeneity and enables effective assessment, even without explicit supervision for these attributes during pretraining.

**Generalization to real-world data.** We further evaluate the model's ability to generalize to real-world settings using the Web dataset. Since ground-truth worker abilities and task difficulties are unavailable, we use individual worker accuracy and task error rate as empirical proxies. As shown in Figure 4, the predictions from CrowdFM trained solely on synthetic data exhibit strong correlation with these observed metrics. This indicates that the model successfully generalizes to real-world data, capturing realistic patterns of worker behavior and task properties.

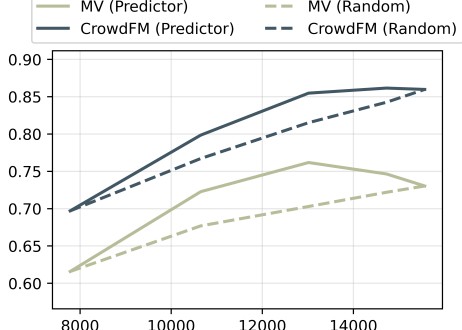

Figure 5: Comparison of CrowdFM and MV under compatibility prediction-based (solid) and random (dashed) assignment strategies over four rounds on the Web dataset. The x-axis shows the number of assigned worker-task pairs, and the y-axis is aggregation accuracy. The leftmost point is the initial 50% observed assignments, and the rightmost is all eligible pairs assigned, which are identical across strategies and therefore overlap.

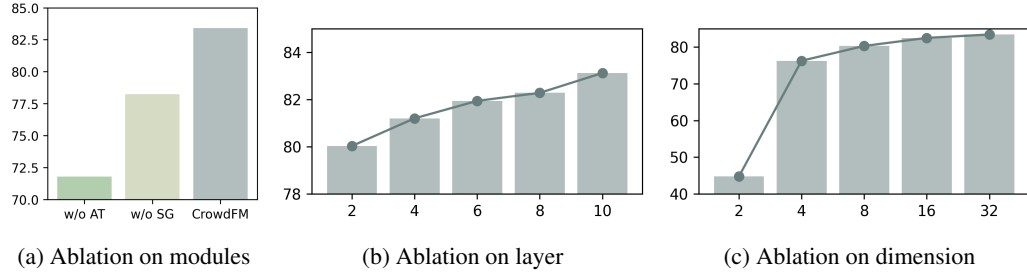

(a) Ablation on modules      (b) Ablation on layer      (c) Ablation on dimension

Figure 6: Ablation study on key modules and hyperparameters of CrowdFM. (a) Effect of attention-based message passing (AT) and the synthetic data generator (SG) on performance. (b) Performance with varying numbers of GNN layers. (c) Performance across different embedding dimensions. The y-axis shows average accuracy over real-world datasets.

### 4.3.2 TASK ASSIGNMENT

A major challenge in crowdsourcing is to maximize annotation quality under limited budgets, ideally by assigning each task to workers most likely to provide correct responses (Ho & Vaughan, 2012). In this section, we demonstrate that CrowdFM, once pretrained, can be effectively repurposed to predict worker–task compatibility, enabling intelligent task assignment.

**Adaptation setup.** We apply a lightweight compatibility head $g_c$ to the model's output embeddings. To enhance discrimination among workers for the same task and prevent bias, we perform data filtering: for each task $t_j$, we sample an equal number of correct and incorrect responses based on agreement with the ground truth $y_j$, ensuring balanced and informative training pairs. The compatibility head is trained with a binary cross-entropy (BCE) objective to predict whether a worker will provide the correct label for a given task:

$$\hat{c}_{ij} = g_c(z_{w_i}, z_{t_j}), \quad \mathcal{L}_c = -\frac{1}{|\mathcal{C}|} \sum_{(i,j) \in |\mathcal{C}|} \log \sigma \left( \hat{c}_{ij} \cdot I(a_{ij}, y_j) \right), \tag{14}$$

where $I(a_{ij}, y_j) = 1$ if $a_{ij} = y_j$, and $-1$ otherwise. $\mathcal{C}$ is the set of filtered worker-task pairs used for training, and $\sigma(\cdot)$ denotes the sigmoid function.

**Evaluation setup.** We design a controlled evaluation protocol to compare task assignment based on predicted compatibility against a random baseline. The evaluation begins with 50% of the initial worker–task assignments used as historical data. CrowdFM leverages these observed assignments to predict compatibilities for the remaining unassigned worker–task pairs, providing guidance for subsequent allocation decisions.

The allocation is carried out over four sequential rounds. In each round, every task is assigned a fixed number of new labeling slots. Under the Predictor strategy, each task selects the top-ranked unassigned workers based on predicted compatibility scores, whereas under the Random strategy, workers are chosen uniformly at random to fill the slots.

**Performance on real-world data.** As shown in Figure 5, using compatibility-based assignment strategy (Predictor) results in significantly higher accuracy for both MV and CrowdFM compared to random assignment (Random). This demonstrates that CrowdFM supports intelligent task assignment in a retraining-free manner, effectively leveraging its pretrained knowledge to guide worker-task allocation.

Notably, the accuracy of MV begins to decline in the third and fourth rounds under the compatibility predictor strategy. This is because higher-quality worker-task pairs are typically allocated in earlier rounds based on the model's predictions. As the process progresses, the remaining pool consists of increasingly noisy assignments, involving more ambiguous or less compatible interactions. While MV is sensitive to this degradation in annotation quality, CrowdFM maintains stable performance, demonstrating its resilience to such challenging cases and highlighting its advantage in handling noisy, real-world annotation patterns.

### 4.4 ABLATION STUDY

**Ablation on modules**  We create two variants of CrowdFM: one replaces the attention mechanism with a mean aggregator (w/o AT), and the other uses a uniformly random generator instead of our synthetic data generator (w/o SG), inspired by (Wu et al., 2023b) (Algorithm 2). As shown in Figure 6a, removing either component degrades performance significantly. The w/o AT variant causes the largest drop in accuracy, confirming that attention is crucial for modeling annotation heterogeneity. The performance drop with w/o SG underscores the importance of diverse synthetic data for sim-to-real transfer.

**Ablation on hyperparameters**  We evaluate the impact of GNN depth $L$ and embedding dimension $d$. As shown in Figure 6b, performance improves steadily with deeper layers, suggesting that longer-range message passing helps capture annotation patterns. In Figure 6c, we observe that dimension 2 is insufficient to represent annotator behaviors, dimension 4 provides basic capacity, and higher dimensions lead to consistent improvements. These trends highlight the scalability of our design and suggest further gains with larger configurations.

## 5 RELATED WORK

**Label Aggregation.**  Existing label aggregation methods typically estimate dataset-specific parameters to model annotation heterogeneous characteristics. PM (Aydin et al., 2014) estimates worker quality, while CATD (Li et al., 2014) quantifies reliability with confidence intervals. GLAD models worker expertise and task difficulty (Whitehill et al., 2009). DS (Dawid & Skene, 1979) and IBCC model per-worker confusion matrices, and EBCC (Li et al., 2019b) extends IBCC by capturing worker correlations. BWA (Li et al., 2019a) adopts inverse-Gamma priors for label estimation. Deep learning approaches such as LAA (Li'ang Yin et al., 2017), TiReMGE (Wu et al., 2023a), and GOVERN (Liu et al., 2024) learn worker and task embeddings through self-supervised learning. Despite their complexity, all these methods require training from scratch on each dataset and lack transferability. In contrast, the simplest method MV enables retraining-free inference across datasets but fails to capture annotation heterogeneity.

HyperLM (Wu et al., 2023b) shares the retraining-free objective for programmatic weak supervision aggregation but is not a foundation model for crowdsourcing with human annotations: HyperLM adopts a uniform random data generation process, which misaligns with real-world annotation patterns; its node-per-binary-annotation graph design incurs high computational cost and does not scale to large-scale annotations or categories; and it lacks explicit representations of workers or tasks, preventing direct application to downstream adaptation.

**Foundation Model.**  Recently, foundation models pretrained on large-scale corpora have demonstrated remarkable generalization capabilities (Zhou et al., 2024). Notably, large language models (LLMs) (Achiam et al., 2023; Guo et al., 2025) excel in natural language understanding and generation, leveraging in-context learning for zero-shot or few-shot adaptation. Concurrently, graph foundation models (Liu et al., 2025; Yu et al., 2025) learn transferable structural representations, enabling effective generalization across graph-based tasks. However, existing foundation models are not designed for crowdsourced sceneries. LLMs focus on sequential text and cannot model worker-task relationships or annotation quality (Appendix I). Graph models rely on rich node features, which are often unavailable in crowd settings. Therefore, these models are fundamentally ill-suited for direct application in crowdsourced label aggregation.

## 6 CONCLUSION

In this work, we introduce CrowdFM, a foundation model for crowdsourced label aggregation. By pretraining a GNN on a wide range of simulated labeling scenarios, it learns to capture annotation heterogeneity and generalizes across diverse datasets without dataset-specific training. The model demonstrates strong cross-dataset generalization, robustness, and adaptability to multiple downstream tasks. We hope this work provides a useful foundation for future research, particularly in improving the realism of synthetic data generation and extending the framework to more complex annotation types such as continuous or structured labels.

## ETHICS STATEMENT

Our method uses only synthetic data for training and publicly available datasets for evaluation. It does not involve human subjects or sensitive information.

## REPRODUCIBILITY STATEMENT

Implementation details and dataset sources are provided in the appendices. Our code and trained models have been open-sourced to ensure reproducibility.

## ACKNOWLEDGMENT

This work was supported in part by the National Natural Science Foundation of China under Grant 62472176, Grant 62172275, Grant 62472277, Grant 62572309 and Grant 62441225, in part by Shanghai East Talents Program under Grant 2023-177, and in part by Huawei Technologies Company, Ltd. under Grant TC20251104033.

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

APPENDIX

## A   THE USE OF LARGE LANGUAGE MODELS (LLMS)

We used GitHub Copilot in Visual Studio Code, powered by models GPT-5 mini and Claude Sonnet 4, to assist with code completion and implementation. The generated code was reviewed, modified, and approved by the authors.

We also used GPT-5 and Qwen3-235B-A22B-2507 to improve the clarity and fluency of the manuscript during the writing process. All text was written and revised by the authors, with LLMs used only for language refinement.

## B   SYNTHETIC DATA GENERATOR IMPLEMENTATION

This section provides implementation details of the synthetic crowdsourced data generator described in Section 3.1. The generator implements the procedure outlined in Algorithm 1 to simulate diverse and realistic crowdsourcing environments through domain randomization.

---

**Algorithm 1** Synthetic Crowdsourced Data Generator

---

1: **Input:** Sample ranges $\mathcal{R}_{\text{size}}, \mathcal{R}_{\text{worker}}, \mathcal{R}_{\text{task}}, \mathcal{R}_{\text{assignment}}$.
2: Sample number of tasks $N$, workers $M$, options $K$, and expected answers per task $A$ from $\mathcal{R}_{\text{size}}$.
3: Sample worker distribution parameters $\mu_\theta, \sigma_\theta$ from $\mathcal{R}_{\text{worker}}$.
4: **for** $i = 1, \ldots, M$ **do**
5:     Sample ability $\theta_i \sim \mathcal{N}(\mu_\theta, \sigma_\theta^2)$.
6: **end for**
7: Sample task distribution parameters $\mu_\beta, \sigma_\beta, \alpha_{\min}, \alpha_{\max}, c_{\text{upper}}$ from $\mathcal{R}_{\text{task}}$.
8: **for** $j = 1, \ldots, N$ **do**
9:     Sample truth label $y_j \sim \text{Uniform}\{1, \ldots, K\}$.
10:     Sample difficulty $\beta_j \sim \mathcal{N}(\mu_\beta, \sigma_\beta^2)$.
11:     Sample discrimination $\alpha_j \sim \text{Uniform}(\alpha_{\min}, \alpha_{\max})$.
12:     Sample guessing $c_j \sim \text{Uniform}(1/K, c_{\text{upper}})$.
13: **end for**
14: Sample assignment parameter $a$ from $\mathcal{R}_{\text{assignment}}$.
15: Draw worker loads $L_i \sim \text{Pareto}(a) + 1$, normalize so $\sum_i L_i \approx N \cdot A$.
16: **for** each task $j$ **do**
17:     Sample required answers $n_j \sim \text{Poisson}(A)$, clip to $[1, M]$.
18:     Select $n_j$ available workers with $L_i > 0$ uniformly without replacement, and decrement their loads.
19: **end for**
20: **for** each assigned pair $(i, j)$ **do**
21:     Compute $p_{ij}$ using Eq. ( 3).
22:     With probability $p_{ij}$, set $a_{ij} = y_j$; otherwise sample a random incorrect option.
23: **end for**
24: **Return** annotation dataset $\mathcal{D} = (\mathcal{A}, \mathcal{Y})$, where $\mathcal{A} = \{(w_i, t_j, a_{ij})\}$ and $\mathcal{Y} = \{y_j\}$.

---

### B.1   SAMPLING RANGES

To promote structural and behavioral diversity across synthetic datasets, all key parameters are sampled from broad, realistic ranges. These ranges are summarized in Table 1.

### B.2   TASK ASSIGNMENT

Each worker $w_i$ is assigned a capacity $L_i$ to reflect heterogeneous participation. The capacities are generated as:

$$L_i = \text{Pareto}(a) + 1, \tag{15}$$

where the shape parameter $a \sim \text{Uniform}(0.2, 1.2)$ is sampled per dataset. The resulting loads are normalized so that $\sum_i L_i \approx N \cdot A$, ensuring alignment between total labeling demand and supply.

| Component | Parameter | Sampling Range |
|---|---|---|
| Dataset Size | Number of tasks $N$ | $[100, 300]$ |
| | Number of workers $M$ | $[20, 100]$ |
| | Number of answer options $K$ | $[2, 20]$ |
| | Expected annotations per task $A$ | $[2, 10]$ |
| Worker Ability | Mean $\mu_\theta$ of $\theta_i \sim \mathcal{N}(\mu_\theta, \sigma_\theta^2)$ | $[-3.0, 1.5]$ |
| | Std $\sigma_\theta$ | $[0.2, 1.2]$ |
| Task Difficulty | Mean $\mu_\beta$ of $\beta_j \sim \mathcal{N}(\mu_\beta, \sigma_\beta^2)$ | $[-2.0, 3.0]$ |
| | Std $\sigma_\beta$ | $[0.5, 1.5]$ |
| Task Discrimination | Lower bound $\alpha_{\min}$ | $[0.5, 1.0]$ |
| | Upper bound $\alpha_{\max}$ | $[1.0, 2.0]$ |
| | $\alpha_j \sim \text{Uniform}(\alpha_{\min}, \alpha_{\max})$ | |
| Task Guessing Rate | Upper bound $c_{\text{upper}}$ | $[1/K, 0.1 + 1/K]$ |
| | $c_j \sim \text{Uniform}(1/K, c_{\text{upper}})$ | |
| Task Assignment | Pareto shape parameter $a$ for worker loads | $[0.2, 1.2]$ |
| Answer Generation | Scaling constant $D$ in 3PL model | 1.7 |

Table 1: Parameter ranges used in the synthetic data generator.

For each task $t_j$, the number of assigned workers $n_j$ is drawn from a Poisson distribution with mean $A$, then clipped to $[1, M]$:

$$n_j = \text{clip}\big(\text{Poisson}(A),\ 1,\ M\big). \tag{16}$$

Workers are selected at random without replacement from those with remaining load ($L_i > 0$), and their loads are decremented after assignment.

## C EXPERIMENTAL ENVIRONMENT

Our model is trained on a machine equipped with an Intel Xeon W-3175X CPU (56 cores, 4.3 GHz), an NVIDIA A100 40GB GPU, and 96 GB RAM. For evaluation, all experiments are conducted on the same machine equipped with an AMD Ryzen 9 9950X processor (32 threads, 4.291 GHz) and 32 GB RAM. Results are averaged over five independent runs with random seeds 42–46 to ensure reproducibility and reliable comparison.

## D   REAL-WORLD CROWDSOURCED DATASETS INFORMATION

We evaluate our method on 22 real-world crowdsourced datasets collected from popular platforms such as Amazon Mechanical Turk. These datasets span multiple domains, including text (e.g., textual entailment), visual (e.g., image classification of birds), and audio (e.g., music genre classification), covering a wide range of modalities and task types.

The datasets include: *Adult* (Mason & Suri, 2012), *Aircrowd6* (Zhang et al., 2015), *Bird* (Welinder et al., 2010), *Dog* (Mozafari et al., 2014), *Face* (Mozafari et al., 2014), *LabelMe* (Rodrigues & Pereira, 2018), *MS* (Rodrigues et al., 2013), *PosSent* (Zheng et al., 2017), *Product* (Wang et al., 2012), *RTE* Snow et al. (2008), *Trec* (Li et al., 2019a), *WS* (Wu et al., 2023a), *Web* (Zhou et al., 2012), *CF*, *CF\**, *SP*, *SP\** Venanzi et al. (2014), *Fact*, *Senti* (Josephy et al., 2014), *ZC_all*, *ZC_in* and *ZC_us* (Demartini et al., 2012). The detailed statistics of these datasets are summarized in Table 2.

While the synthetic data generator samples key parameters from broad ranges (Table 1), several real-world datasets fall outside these ranges in terms of size and annotation density. For example, *Fact* and *Senti* contain 42,624 and 98,980 tasks, respectively, far exceeding the synthetic range of $[100, 300]$. Datasets such as *Bird*, *CF\**, *PosSent*, *SP\**, and *WS* exhibit very high annotation densities per task, compared to the synthetic range of $[2, 10]$. These extreme configurations were not included in the synthetic generator to maintain computational efficiency and avoid overly sparse or dense settings that are uncommon in typical crowdsourcing scenarios. Despite these differences, CrowdFM maintains strong performance, demonstrating robustness to variations in dataset scale and annotation density that differ substantially from the synthetic training distributions.

| Dataset | #Worker | #Task | #Option | #Answer | #Answer/#Worker | #Answer/#Task |
|---------|---------|-------|---------|---------|------------------|----------------|
| Adult | 825 | 11,040 | 4 | 92,721 | 112.39 | 8.40 |
| Aircrowd6 | 51 | 593 | 6 | 1,588 | 31.14 | 2.68 |
| Bird | 39 | 108 | 2 | 4,212 | 108.00 | 39.00 |
| CF | 461 | 300 | 5 | 1,720 | 3.73 | 5.73 |
| CF* | 110 | 300 | 5 | 6,030 | 54.82 | 20.10 |
| Dog | 109 | 807 | 4 | 8,070 | 74.04 | 10.00 |
| Face | 27 | 584 | 4 | 5,242 | 194.15 | 8.98 |
| Fact | 57 | 42,624 | 3 | 216,725 | 3802.19 | 5.08 |
| LabelMe | 59 | 1,000 | 8 | 2,547 | 43.17 | 2.55 |
| MS | 44 | 700 | 10 | 2,945 | 66.93 | 4.21 |
| PosSent | 85 | 1,000 | 2 | 20,000 | 235.29 | 20.00 |
| Product | 176 | 8,315 | 2 | 24,945 | 141.73 | 3.00 |
| RTE | 164 | 800 | 2 | 8,000 | 48.78 | 10.00 |
| SP | 203 | 4,999 | 2 | 27,746 | 136.68 | 5.55 |
| SP* | 143 | 500 | 2 | 10,000 | 69.93 | 20.00 |
| Senti | 1,960 | 98,980 | 5 | 569,375 | 290.50 | 5.75 |
| Trec | 762 | 19,033 | 2 | 88,385 | 115.99 | 4.64 |
| WS | 111 | 300 | 5 | 6,000 | 54.05 | 20.00 |
| Web | 177 | 2,665 | 5 | 15,567 | 87.95 | 5.84 |
| ZC_all | 78 | 2,040 | 2 | 21,855 | 280.19 | 10.71 |
| ZC_in | 25 | 2,040 | 2 | 11,205 | 448.20 | 5.49 |
| ZC_us | 74 | 2,040 | 2 | 12,190 | 164.73 | 5.98 |

Table 2: Summary statistics of 22 real-world crowdsourcing datasets: number of workers, number of tasks, number of answer options, total number of answers, average answers per worker, and average answers per task.

# E   PER-DATASET EXPERIMENTAL RESULT

## E.1   ACCURACY RESULT

| | Adult | Aircrowd6 | Bird | CF | CF* | Dog | Face | Fact |
|---|---|---|---|---|---|---|---|---|
| MV | 75.98 | 80.84 | 75.93 | 87.87 | 85.27 | 81.83 | 63.90 | 90.17 |
| PM | 76.58 | 76.12 | 78.70 | 88.53 | 85.67 | 81.91 | 59.76 | 89.58 |
| CATD | 74.83 | 80.37 | 77.78 | 88.13 | 86.00 | 82.01 | 61.64 | 88.89 |
| DS | 73.57 | 84.49 | 87.96 | 79.67 | 85.67 | 84.26 | 64.04 | 84.90 |
| BWA | 74.17 | 82.46 | 75.93 | 89.33 | 86.00 | 83.15 | 61.82 | 88.72 |
| IBCC | 74.47 | 77.07 | 88.89 | 88.33 | 85.67 | 83.89 | 64.04 | 87.67 |
| EBCC | 74.59 | 81.45 | 86.30 | 88.00 | 86.07 | 84.06 | 62.71 | 89.03 |
| GLAD | 76.58 | 80.61 | 76.85 | 88.00 | 85.67 | 83.52 | 62.84 | 90.17 |
| LAA | 94.28 | 77.34 | 92.22 | 83.87 | 85.60 | 84.34 | 66.06 | 61.44 |
| TiReMGE | 75.26 | 78.75 | 55.56 | 87.87 | 84.53 | 81.14 | 63.29 | 90.17 |
| GOVERN | 70.51 | 81.69 | 87.22 | 90.07 | 85.80 | 82.48 | 66.13 | NaN |
| HyperLM | 78.68 | 82.29 | 80.56 | 86.67 | 83.00 | 83.27 | 66.78 | 89.58 |
| CrowdFM | 76.16 | 81.92 | 79.63 | 88.07 | 85.87 | 82.26 | 64.14 | 90.21 |

| | LabelMe | MS | PosSent | Product | RTE | SP | SP* | Senti |
|---|---|---|---|---|---|---|---|---|
| MV | 76.64 | 70.37 | 93.60 | 89.66 | 89.35 | 88.62 | 94.16 | 88.26 |
| PM | 73.76 | 79.51 | 95.08 | 89.81 | 92.12 | 89.65 | 94.20 | 83.30 |
| CATD | 76.60 | 79.11 | 95.50 | 92.66 | 92.45 | 91.45 | 94.36 | 88.32 |
| DS | 79.40 | 76.57 | 96.00 | 93.99 | 92.75 | 91.50 | 94.40 | 81.50 |
| BWA | 77.40 | 78.57 | 95.60 | 91.94 | 92.75 | 91.70 | 94.60 | 89.00 |
| IBCC | 76.40 | 79.00 | 96.00 | 93.83 | 92.75 | 91.50 | 94.40 | 83.10 |
| EBCC | 78.60 | 78.80 | 95.70 | 93.50 | 93.00 | 91.22 | 94.16 | 86.24 |
| GLAD | 76.70 | 78.43 | 95.20 | 92.15 | 92.62 | 91.78 | 94.80 | 89.40 |
| LAA | 62.08 | 68.03 | 95.84 | 75.27 | 91.75 | 88.51 | 94.56 | NaN |
| TiReMGE | 75.92 | 71.60 | 93.54 | 89.66 | 91.68 | 88.35 | 94.36 | 88.84 |
| GOVERN | 77.86 | 76.11 | 95.48 | 85.26 | 93.15 | 90.07 | 94.24 | NaN |
| HyperLM | 77.70 | 70.57 | 93.20 | 67.05 | 91.87 | 88.96 | 94.40 | 85.60 |
| CrowdFM | 77.12 | 79.80 | 95.08 | 89.93 | 91.53 | 89.92 | 94.36 | 88.18 |

| | Trec | WS | Web | ZC_all | ZC_in | ZC_us |
|---|---|---|---|---|---|---|
| MV | 65.10 | 85.27 | 73.03 | 83.11 | 74.02 | 86.08 |
| PM | 69.37 | 85.73 | 39.40 | 81.75 | 73.55 | 81.81 |
| CATD | 60.79 | 86.00 | 80.35 | 82.22 | 76.91 | 90.88 |
| DS | 70.33 | 85.67 | 82.21 | 79.61 | 75.74 | 82.25 |
| BWA | 60.44 | 86.00 | 82.25 | 83.48 | 76.37 | 91.08 |
| IBCC | 70.55 | 85.67 | 75.08 | 79.51 | 76.96 | 82.70 |
| EBCC | 70.37 | 85.93 | 74.36 | 86.25 | 78.09 | 91.23 |
| GLAD | 57.49 | 85.67 | 79.68 | 81.52 | 77.01 | 83.82 |
| LAA | 18.99 | 85.60 | 87.45 | 79.39 | 72.07 | 82.12 |
| TiReMGE | 64.49 | 84.47 | 58.98 | 83.01 | 77.97 | 87.01 |
| GOVERN | 65.03 | 85.00 | 91.05 | 78.98 | 73.03 | 83.09 |
| HyperLM | 66.15 | 83.33 | 78.97 | 77.11 | 72.25 | 79.90 |
| CrowdFM | 65.14 | 85.87 | 85.96 | 83.38 | 74.06 | 86.53 |

Table 3: Per-method accuracy on each dataset. Colors reflect performance relative to the MV baseline (green: superior, red: inferior).

## E.2 RUNTIME RESULT

|  | Adult | Aircrowd6 | Bird | CF | CF* | Dog | Face | Fact |
|---|---|---|---|---|---|---|---|---|
| MV | 0.05 | 0.01 | 0.00 | 0.01 | 0.01 | 0.01 | 0.01 | 0.14 |
| PM | 0.80 | 0.01 | 0.01 | 0.01 | 0.02 | 0.04 | 0.02 | 2.11 |
| CATD | 4.11 | 0.06 | 0.06 | 0.05 | 0.11 | 0.16 | 0.09 | 12.77 |
| DS | 7.59 | 0.14 | 0.11 | 0.18 | 0.34 | 0.40 | 0.26 | 20.18 |
| BWA | 0.20 | 0.03 | 0.01 | 0.01 | 0.01 | 0.02 | 0.02 | 0.24 |
| IBCC | 0.09 | 0.01 | 0.01 | 0.01 | 0.01 | 0.02 | 0.01 | 0.10 |
| EBCC | 1.20 | 0.07 | 0.01 | 0.11 | 0.02 | 0.04 | 0.06 | 5.27 |
| GLAD | 1276.54 | 2.39 | 9.77 | 10.89 | 19.21 | 41.55 | 42.90 | 2100.37 |
| LAA | 2782.28 | 13.18 | 6.40 | 72.38 | 17.35 | 20.80 | 12.41 | 237.67 |
| TiReMGE | 36.91 | 3.51 | 3.77 | 4.35 | 4.40 | 5.18 | 4.33 | 104.60 |
| GOVERN | 397.52 | 3.51 | 2.29 | 4.30 | 4.00 | 6.59 | 6.13 | NaN |
| HyperLM | 0.73 | 0.02 | 0.01 | 0.02 | 0.04 | 0.04 | 0.02 | 0.58 |
| CrowdFM | 0.99 | 0.03 | 0.04 | 0.03 | 0.06 | 0.07 | 0.10 | 2.25 |

|  | LabelMe | MS | PosSent | Product | RTE | SP | SP* | Senti |
|---|---|---|---|---|---|---|---|---|
| MV | 0.01 | 0.01 | 0.01 | 0.04 | 0.01 | 0.03 | 0.01 | 0.41 |
| PM | 0.02 | 0.01 | 0.09 | 0.20 | 0.04 | 0.19 | 0.04 | 5.48 |
| CATD | 0.08 | 0.07 | 0.33 | 0.75 | 0.15 | 0.66 | 0.17 | 31.55 |
| DS | 0.30 | 0.38 | 0.55 | 1.05 | 0.22 | 0.92 | 0.26 | 74.99 |
| BWA | 0.04 | 0.03 | 0.02 | 0.10 | 0.02 | 0.10 | 0.01 | 0.96 |
| IBCC | 0.07 | 0.02 | 0.02 | 0.10 | 0.01 | 0.06 | 0.01 | 1.76 |
| EBCC | 0.12 | 0.19 | 0.08 | 0.17 | 0.04 | 0.26 | 0.02 | 56.10 |
| GLAD | 12.36 | 81.72 | 75.42 | 195.45 | 38.58 | 227.08 | 42.83 | 4729.94 |
| LAA | 38.94 | 42.59 | 22.20 | 83.68 | 25.71 | 69.29 | 27.23 | NaN |
| TiReMGE | 4.20 | 4.15 | 10.84 | 16.06 | 5.20 | 15.00 | 5.57 | 280.19 |
| GOVERN | 5.00 | 2.37 | 12.30 | 194.46 | 6.81 | 66.77 | 8.59 | NaN |
| HyperLM | 0.04 | 0.06 | 0.03 | 0.08 | 0.02 | 0.07 | 0.02 | 16.72 |
| CrowdFM | 0.04 | 0.04 | 0.18 | 0.22 | 0.07 | 0.23 | 0.09 | 5.75 |

|  | Trec | WS | Web | ZC_all | ZC_in | ZC_us |
|---|---|---|---|---|---|---|
| MV | 0.06 | 0.01 | 0.02 | 0.02 | 0.02 | 0.02 |
| PM | 0.84 | 0.02 | 0.09 | 0.13 | 0.06 | 0.07 |
| CATD | 4.29 | 0.10 | 0.39 | 0.43 | 0.25 | 0.26 |
| DS | 4.65 | 0.33 | 1.10 | 0.64 | 0.35 | 0.39 |
| BWA | 0.20 | 0.01 | 0.05 | 0.04 | 0.03 | 0.05 |
| IBCC | 0.08 | 0.01 | 0.07 | 0.03 | 0.03 | 0.03 |
| EBCC | 0.57 | 0.02 | 0.29 | 0.10 | 0.08 | 0.09 |
| GLAD | 1475.34 | 32.67 | 105.35 | 140.51 | 117.52 | 95.33 |
| LAA | 883.01 | 37.25 | 169.53 | 41.55 | 38.69 | 42.20 |
| TiReMGE | 39.15 | 4.33 | 10.33 | 11.58 | 7.02 | 8.38 |
| GOVERN | 1121.03 | 5.03 | 19.24 | 16.70 | 12.88 | 12.98 |
| HyperLM | 0.56 | 0.04 | 0.10 | 0.03 | 0.02 | 0.03 |
| CrowdFM | 0.93 | 0.06 | 0.14 | 0.17 | 0.10 | 0.17 |

Table 4: Per-method runtime on each dataset, measured in seconds..

# F  SIM-TO-REAL GAP

Since the real-world datasets do not provide attributes such as worker quality, task difficulty, discrimination or guessing parameters defined in our synthetic data generation procedure, we sample 10,000 synthetic datasets and compute 16 metrics for comparison with real-world datasets. Specifically, we compare accuracy and load ratios (actual annotation number / available number) between synthetic and real data, reporting the mean, standard deviation, 25th percentile (Q25), and 75th percentile (Q75) for both workers and tasks.

The results are shown in Figure 1 to Figure 4. We visualize the distributions using boxplots overlaid with individual data points representing each dataset. As can be observed, the synthetic data covers the majority of real datasets and exhibits a similar distribution pattern. We compute the Wasserstein distance between synthetic and real distributions, all of which are below 0.15, indicating close alignment. Notably, some real datasets such as Senti, Fact, Adult, and Trec are outliers that deviate from the overall pattern. Nevertheless, our method achieves performance on these datasets that is either superior to or comparable with MV. In contrast, most other methods underperform MV on these datasets, as demonstrated in Table 3.

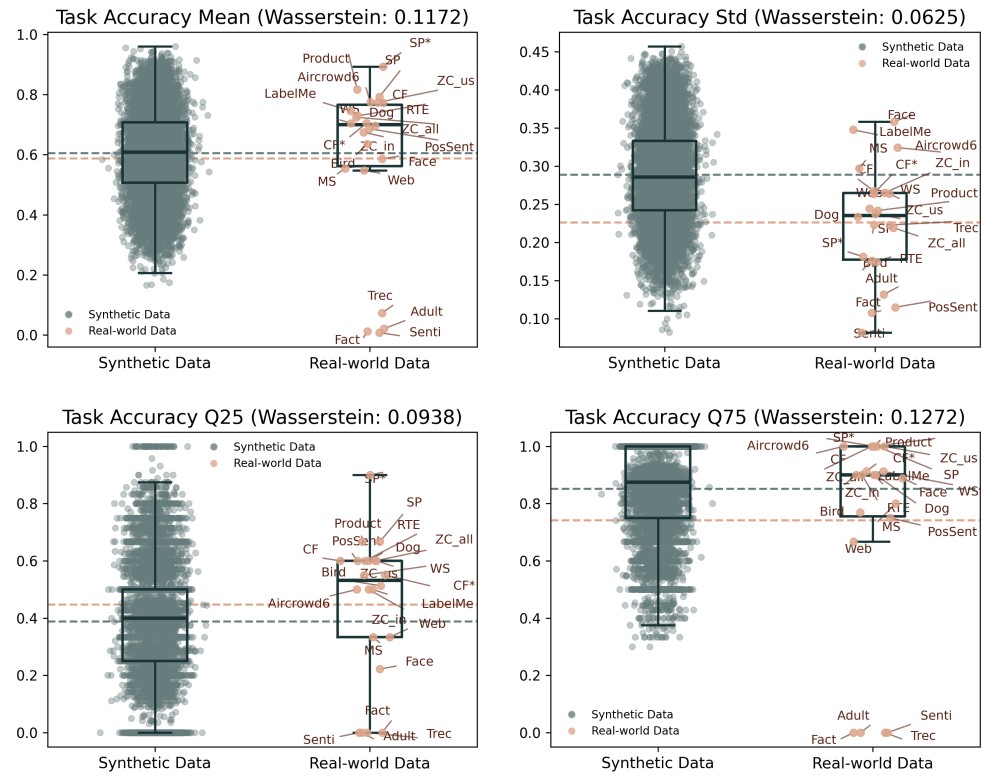

Figure 1: Comparison of task accuracy distributions.

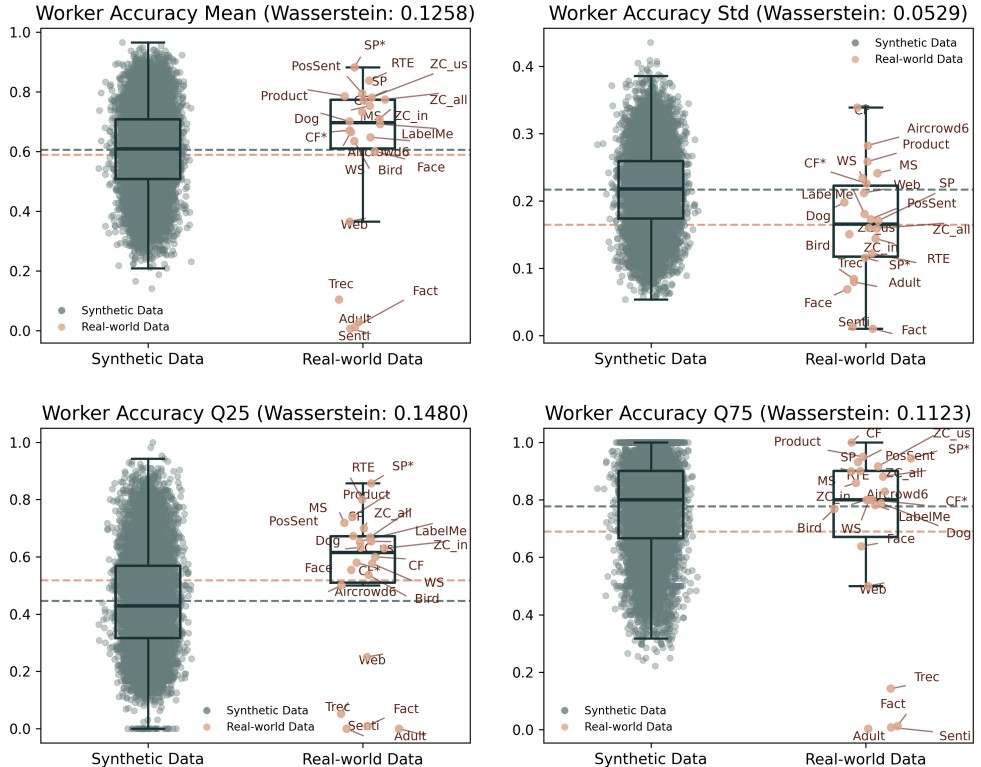

Figure 2: Comparison of worker accuracy distributions.

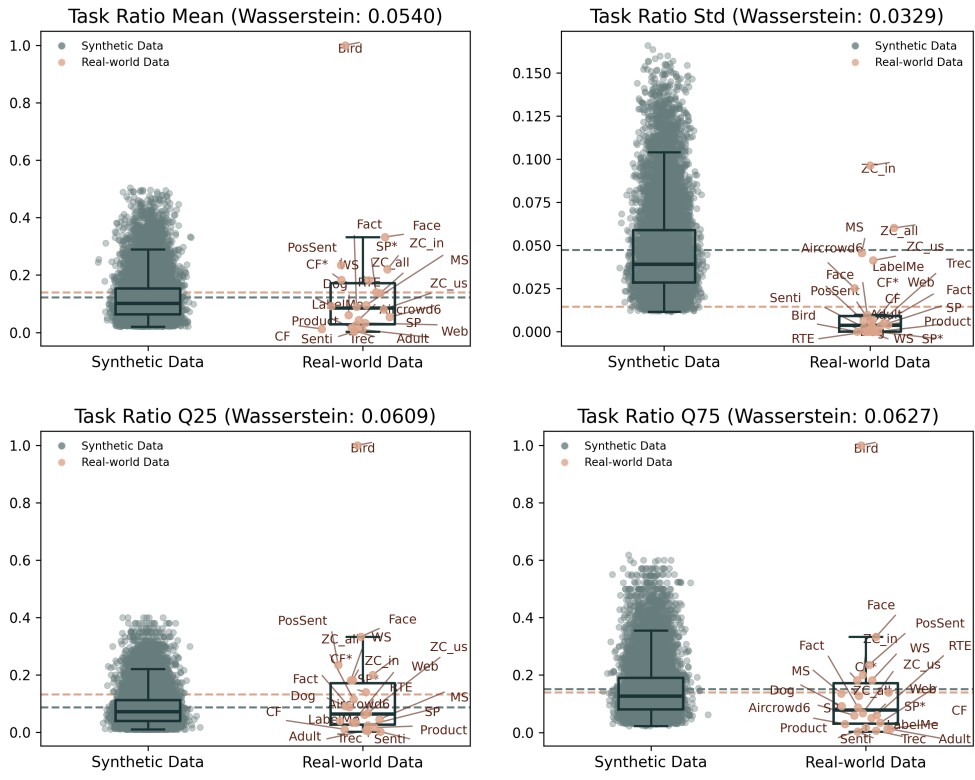

Figure 3: Comparison of task load ratios.

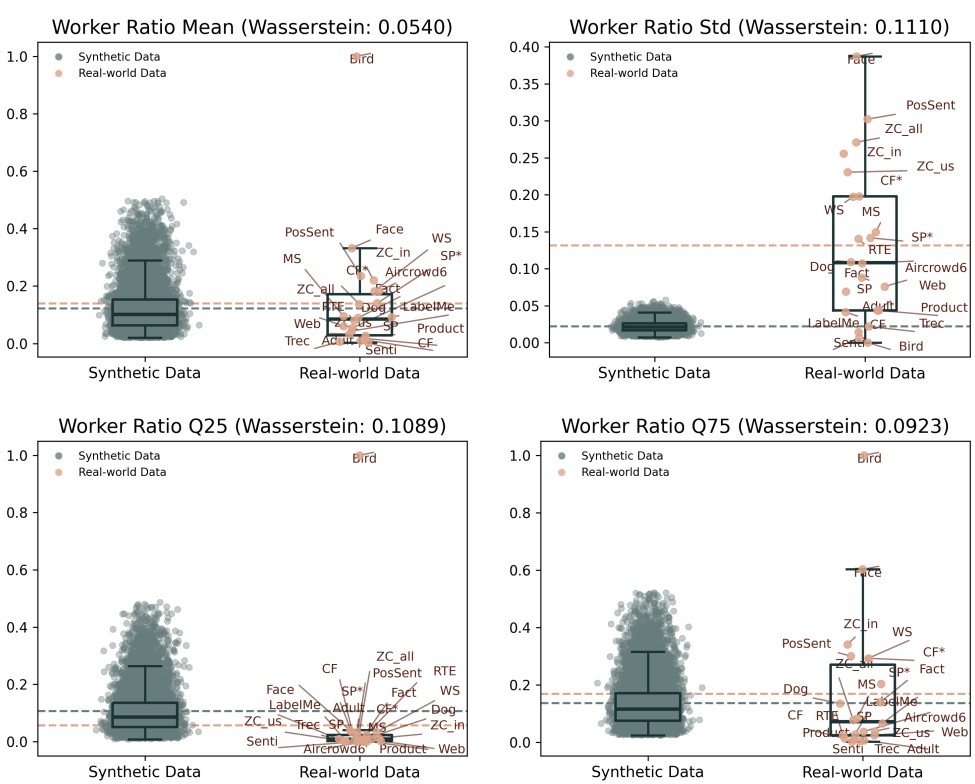

Figure 4: Comparison of worker load ratios.

# G    PERFORMANCE ON SYNTHETIC DATA

To evaluate the robustness and generalization of CrowdFM, we conduct comprehensive experiments on synthetic datasets with systematically varied characteristics. Specifically, we craft 1,000 synthetic datasets, and measure the accuracy improvement of CrowdFM over MV, across five key data properties: worker ability, task difficulty, answer density, category distribution entropy, and number of options. As shown in Figure 5, CrowdFM consistently outperforms MV across on most scenarios, demonstrating its effectiveness under diverse conditions.

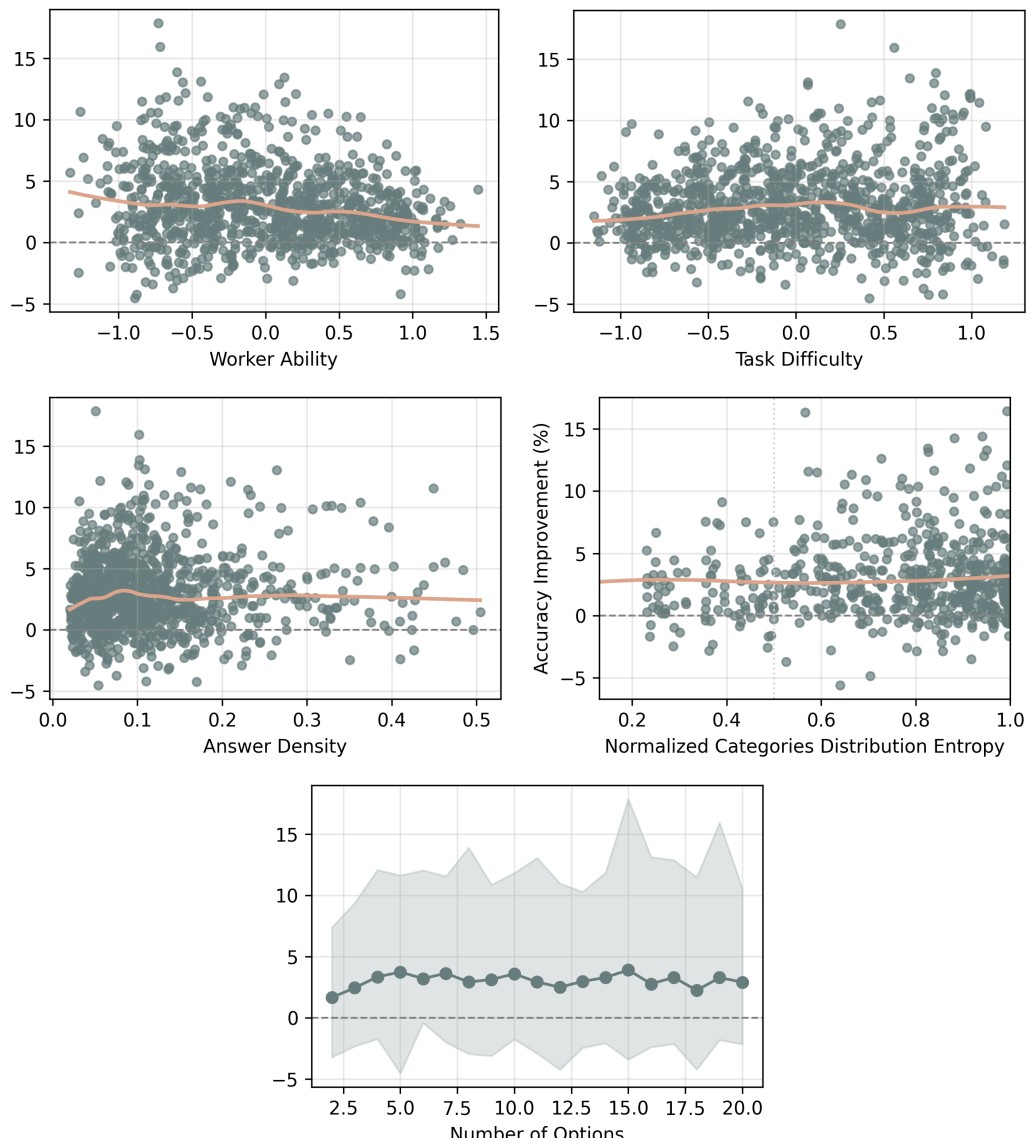

Figure 5: Accuracy improvement of CrowdFM over MV across five synthetic data dimensions. The y-axis in all subplots represents the absolute improvement in classification accuracy, expressed in percentage points. The x-axis in each subplot corresponds to a systematically varied property of the synthetic datasets.

## H   COMPARISON WITH UNIFORMLY SYNTHETIC GENERATION

To investigate the importance of realistic modeling in synthetic data generation, we implement a baseline approach inspired by the uniform generation strategy used in HyperLM Wu et al. (2023b), which assumes that labeling functions start from random guessing and are then calibrated to be better than random. The full procedure is detailed in Algorithm 2.

We generate 10,000 datasets using this uniform synthesis method and compare their statistical properties with real world datasets across the same 16 metrics as in Appendix F, including accuracy and load ratios for workers and tasks (mean, std, Q25, Q75). However, as shown in Figure 6 to Figure 9, the distributions exhibit significant discrepancies from real data. Most Wasserstein distances between synthetic and real distributions range from 0.3 to 0.6 across metrics, substantially higher than those produced by our proposed simulator, indicating a large sim to real gap. Models trained on such synthetic data may learn unrealistic assumptions such as independent errors or uniform reliability, leading to poor real world generalization.

---

**Algorithm 2** Uniformly Synthetic Data Generator

---

1: **Input:** Sample ranges $\mathcal{R}_{\text{size}}, \mathcal{R}_{\text{worker}}, \mathcal{R}_{\text{task}}$.
2: Sample number of tasks $N$, workers $M$, options $K$.
3: Generate ground truth labels $\mathcal{Y} = (y_1, \ldots, y_N)$, where $y_j \sim \text{Uniform}\{1, \ldots, K\}$.
4: **for** $i = 1, \ldots, M$ **do**
5:     Sample number of assigned tasks $k_i \sim \text{Uniform}\{1, N\}$.
6:     Randomly select $k_i$ distinct tasks.
7:     Set $a_{ij} \sim \text{Uniform}\{1, \ldots, K\}$ for the selected $k_i$ tasks.
8:     Count correct predictions $c_i$.
9: **end for**
10: Compute minimum required correct count: $c_{\min} = \lceil N/K \rceil$.
11: Identify well-performing workers: $B = \{i \mid c_i \geq c_{\min}\}$.
12: Determine how many underperforming workers need improvement:
    $\Delta = \max\left(0, \left\lceil \frac{M}{2} \right\rceil - |B|\right)$.
13: **if** $\Delta > 0$ **then**
14:     From workers not in $B$, sort by deficit $(c_{\min} - c_i)$ in descending order.
15:     Select top $\Delta$ workers with largest deficits.
16:     **for** each selected worker $i$ **do**
17:         Correct up to $(c_{\min} - c_i)$ errors by setting $a_{ij} = y_j$ at random wrong positions.
18:     **end for**
19: **end if**
20: **Return** annotation dataset $\mathcal{D} = (\mathcal{A}, \mathcal{Y})$, where $\mathcal{A} = \{(w_i, t_j, a_{ij})\}$ and $\mathcal{Y} = \{y_j\}$.

---

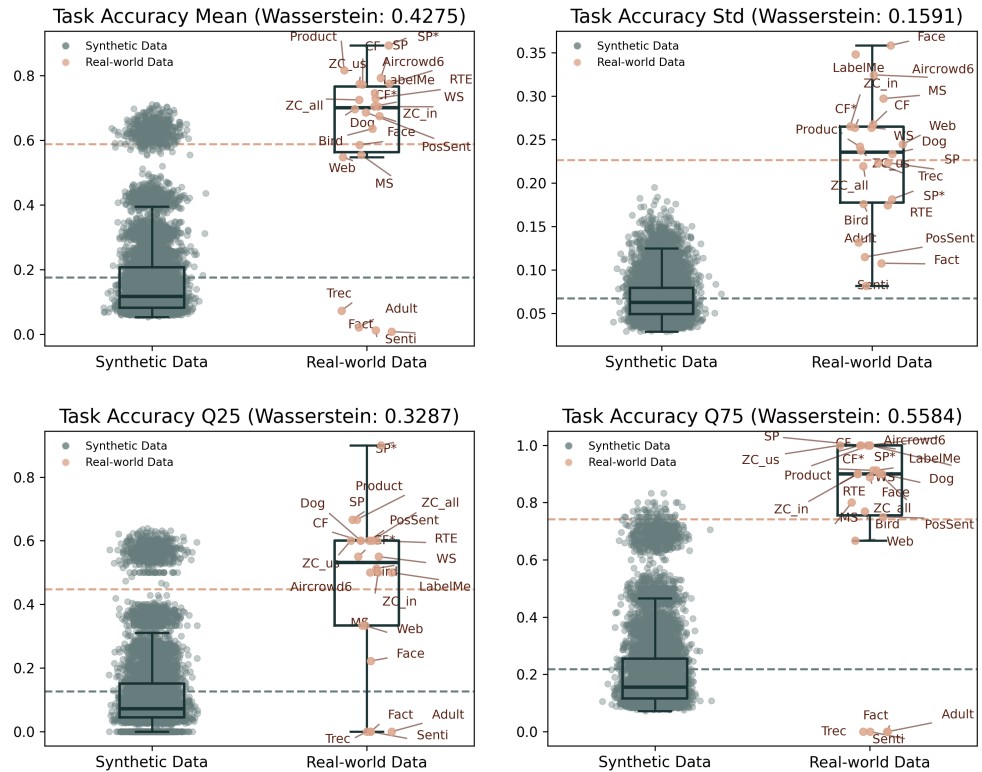

Figure 6: Comparison of task accuracy distributions. (Uniformly synthetic generation)

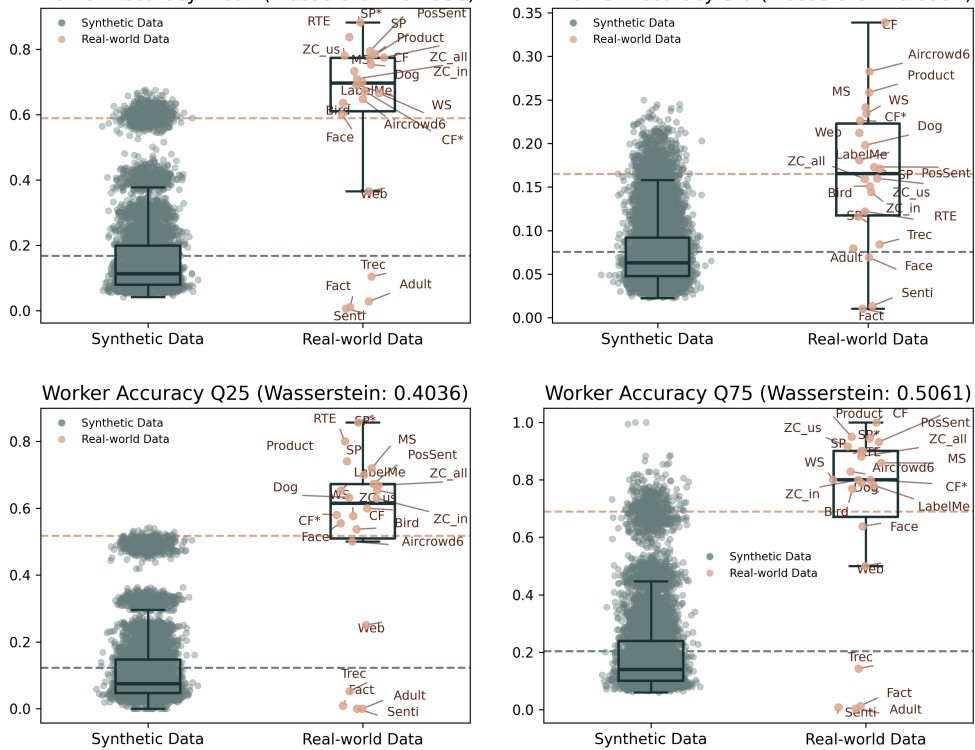

Figure 7: Comparison of worker accuracy distributions. (Uniformly synthetic generation)

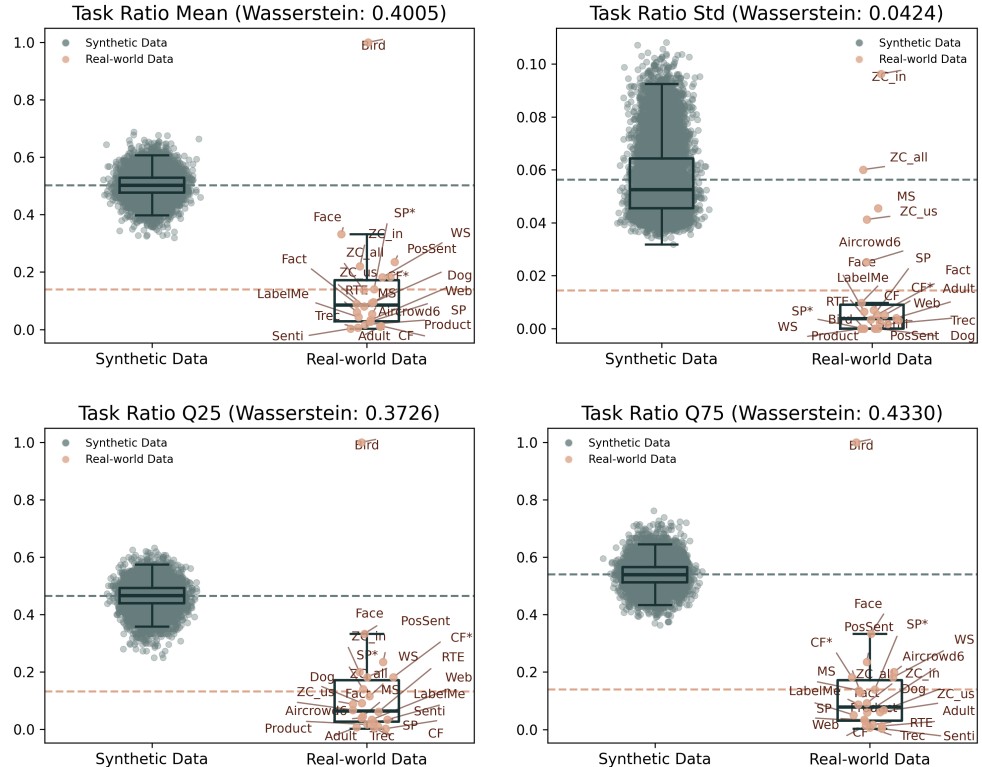

Figure 8: Comparison of task load ratios. (Uniformly synthetic generation)

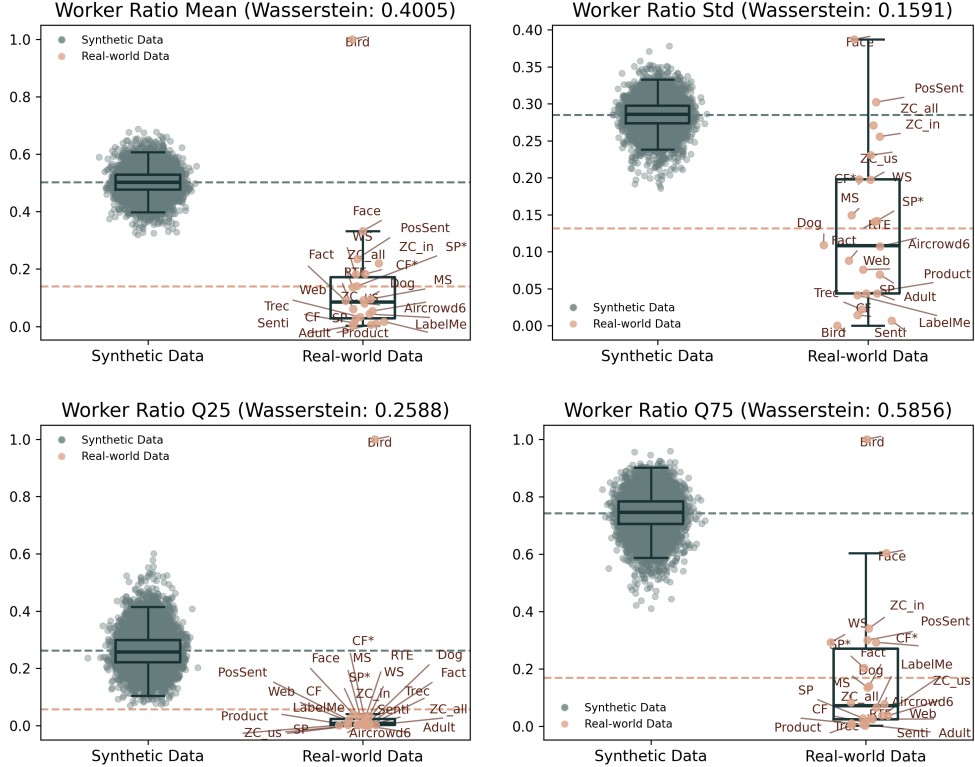

Figure 9: Comparison of worker load ratios. (Uniformly synthetic generation)

# I  LLMs-based Label Aggregation

Most Existing crowdsourcing aggregation methods typically rely on dataset-specific parameter estimation. In this section, we explore an alternative, parameter-estimation-free approach that leverages large language models (LLMs) to perform label aggregation without any dataset-specific training.

In this method, the LLM is provided with a CSV file of raw annotations (`label.csv`) and prompts it to generate the predicted aggregated truth CSV (`predict.csv`) in a specified format. The input prompt used to guide the LLM is as follows:

---

**LLM Label Aggregation Prompt**

You are a precise label aggregation engine for crowdsourced classification tasks.
Input format:
- task: integer ID of the task
- worker: integer ID of the annotator
- answer: annotation label

Your job:
1. Analyze the annotations and infer the most likely true label for each task.
2. Consider factors such as worker reliability, task difficulty, and potential biases when estimating the truth.
3. Output the aggregated results in CSV format with two columns: 'task' and 'truth'.
4. Only include tasks that appear in the input.

Do not include any explanations, headers beyond 'task,truth', or additional text.
Only output the CSV data starting with:
task,truth

Input:
⟨⟨label.csv⟩⟩

Output:

---

We selected three representative LLMs for evaluation: GROK 4 FAST (xAI, 2025), NEMOTRON NANO 9B V2 (NVIDIA, 2025), and DEEPSEEK V3.1 (DeepSeek-AI, 2024), with the results presented in Table 5.

Overall, the performance of LLM-based aggregation is generally poor. Only in a few datasets (e.g., PosSent, SP*, RTE) do some models achieve accuracies close to MV, while on the majority of datasets, all LLMs perform substantially worse. In many cases, accuracy drops dramatically, and some models even fail to produce meaningful outputs.

More critically, LLMs consume a very large number of tokens during inference. This excessive token usage not only leads to significant computational and monetary costs but also imposes a strict scalability bottleneck, preventing the method from running on larger datasets.

In summary, while LLMs are foundation models that demonstrate strong capabilities across a wide range of scenarios, their application to label aggregation is still limited. They struggle in both effectiveness and efficiency: they cannot reliably capture structured information, account for annotation heterogeneity, or infer aggregated labels without incurring excessive computational costs.

| | MV | GROK 4 FAST | NEMOTRON NANO 9B V2 | DEEPSEEK V3.1 |
|---|---|---|---|---|
| **Adult** | | | | |
| Accuracy | 75.98 | NaN | NaN | NaN |
| Prompt Tokens | - | NaN | NaN | NaN |
| Completion Tokens | - | NaN | NaN | NaN |
| **Aircrowd6** | | | | |
| Accuracy | 80.84 | 29.51 | 24.62 | 29.51 |

|  | MV | GROK 4 FAST | NEMOTRON NANO 9B V2 | DEEPSEEK V3.1 |
|---|---|---|---|---|
| Prompt Tokens | - | 9,797 | 12,945 | 9,797 |
| Completion Tokens | - | 15,111 | 6,151 | 15,111 |
| **Bird** | | | | |
| Accuracy | 75.93 | 56.48 | 44.44 | 75.00 |
| Prompt Tokens | - | 25,541 | 32,709 | 25,428 |
| Completion Tokens | - | 5,153 | 2,240 | 436 |
| **CF** | | | | |
| Accuracy | 87.87 | 38.33 | 32.00 | 38.00 |
| Prompt Tokens | - | 10,589 | 16,369 | 10,476 |
| Completion Tokens | - | 14,550 | 3,025 | 1,204 |
| **CF\*** | | | | |
| Accuracy | 85.27 | 37.67 | 6.00 | 37.33 |
| Prompt Tokens | - | 36,449 | 51,737 | 36,336 |
| Completion Tokens | - | 3,230 | 8,348 | 1,204 |
| **Dog** | | | | |
| Accuracy | 81.83 | 36.18 | 30.24 | 35.81 |
| Prompt Tokens | - | 48,689 | 70,288 | 48,576 |
| Completion Tokens | - | 10,478 | 7,836 | 3,232 |
| **Face** | | | | |
| Accuracy | 63.90 | 46.06 | 30.48 | 46.06 |
| Prompt Tokens | - | 31,721 | 45,325 | 31,608 |
| Completion Tokens | - | 10,930 | 4,322 | 2,340 |
| **Fact** | | | | |
| Accuracy | 90.17 | NaN | NaN | NaN |
| Prompt Tokens | - | NaN | NaN | NaN |
| Completion Tokens | - | NaN | NaN | NaN |
| **LabelMe** | | | | |
| Accuracy | 76.64 | 14.90 | 4.40 | 15.90 |
| Prompt Tokens | - | 15,551 | 22,210 | 15,438 |
| Completion Tokens | - | 9,431 | 7,457 | 4,004 |
| **MS** | | | | |
| Accuracy | 70.37 | 20.29 | 0.00 | 20.43 |
| Prompt Tokens | - | 17,939 | 25,403 | 17,826 |
| Completion Tokens | - | 6,693 | 1,802 | 2,804 |
| **PosSent** | | | | |
| Accuracy | 93.60 | 90.10 | NaN | 84.40 |
| Prompt Tokens | - | 120,269 | NaN | 120,156 |
| Completion Tokens | - | 6,553 | NaN | 4,004 |
| **Product** | | | | |
| Accuracy | 89.66 | 1.84 | NaN | NaN |
| Prompt Tokens | - | 171,884 | NaN | NaN |
| Completion Tokens | - | 3,313 | NaN | NaN |
| **RTE** | | | | |
| Accuracy | 89.25 | 82.87 | 53.25 | 90.62 |
| Prompt Tokens | - | 48,269 | 68,891 | 48,156 |
| Completion Tokens | - | 8,872 | 5,378 | 3,204 |
| **SP** | | | | |
| Accuracy | 88.62 | NaN | NaN | NaN |
| Prompt Tokens | - | NaN | NaN | NaN |
| Completion Tokens | - | NaN | NaN | NaN |
| **SP\*** | | | | |

| | MV | GROK 4 FAST | NEMOTRON NANO 9B V2 | DEEPSEEK V3.1 |
|---|---|---|---|---|
| Accuracy | 94.16 | 91.00 | 79.20 | 93.40 |
| Prompt Tokens | - | 60,269 | 90,201 | 60,156 |
| Completion Tokens | - | 8,863 | 8,767 | 2,000 |
| **Senti** | | | | |
| Accuracy | 88.26 | NaN | NaN | NaN |
| Prompt Tokens | - | NaN | NaN | NaN |
| Completion Tokens | - | NaN | NaN | NaN |
| **Trec** | | | | |
| Accuracy | 65.10 | NaN | NaN | NaN |
| Prompt Tokens | - | NaN | NaN | NaN |
| Completion Tokens | - | NaN | NaN | NaN |
| **WS** | | | | |
| Accuracy | 85.27 | 38.00 | 28.33 | 38.00 |
| Prompt Tokens | - | 36,269 | 50,980 | 36,156 |
| Completion Tokens | - | 2,377 | 4,515 | 1,204 |
| **Web** | | | | |
| Accuracy | 73.03 | 1.93 | NaN | 20.47 |
| Prompt Tokens | - | 103,395 | NaN | 103,282 |
| Completion Tokens | - | 13,766 | NaN | 12,329 |
| **ZC_all** | | | | |
| Accuracy | 83.11 | 6.67 | NaN | 76.86 |
| Prompt Tokens | - | 142,419 | NaN | 142,306 |
| Completion Tokens | - | 4,923 | NaN | 15,364 |
| **ZC_in** | | | | |
| Accuracy | 74.02 | 65.34 | NaN | 59.26 |
| Prompt Tokens | - | 73,154 | NaN | 73,041 |
| Completion Tokens | - | 13,194 | NaN | 7,264 |
| **ZC_us** | | | | |
| Accuracy | 86.08 | 77.45 | 7.06 | 72.99 |
| Prompt Tokens | - | 79,584 | 113,421 | 79,471 |
| Completion Tokens | - | 12,236 | 6,978 | 10,414 |

Table 5: Accuracy and token usage of LLM-based label aggregation compared to MV across datasets (NaN indicates the model could not produce a result).

