# OpenReview forum: "Towards a Foundation Model for Crowdsourced Label Aggregation"
_ICLR.cc/2026/Conference — ICLR 2026 Poster_

### Official Review · Reviewer_QfN2 · 2025-10-25

**Soundness:** 2
**Presentation:** 2
**Contribution:** 1
**Rating:** 2
**Confidence:** 5

**Summary:**

This paper introduces CrowdFM, a graph neural network-based "foundation model" for crowdsourced label aggregation, pretrained on synthetic data to enable retraining-free, cross-dataset generalization. The model is evaluated on 22 real-world datasets and shows strong empirical performance and efficiency. However, the main weakness of this paper is its failure to acknowledge and position itself with respect to highly relevant prior work—specifically, the hyper label model (Wu et al., ICLR 2023). The methodology, motivation, and even the technical design of CrowdFM are very similar to those of the hyper label model, with the only substantive difference being the application domain (crowdsourcing vs. programmatic weak supervision) and the technical details in training data generating and the graph neural network architecture design. The paper overstates its novelty and does not provide a meaningful discussion or comparison to this existing work. As a result, the contribution is incremental and the claims of being the "first" to propose such a foundation model are not justified.

**Strengths:**

- The paper addresses a real and important challenge in crowdsourced label aggregation: the need for scalable, retraining-free aggregation methods that generalize across datasets.
- The paper provides a synthetic data generation process for training such models.
- CrowdFM achieves fast inference times, comparable to simple methods like Majority Voting, while outperforming more complex dataset-specific models.

**Weaknesses:**

1.  Lack of proper acknowledgement of closely related work. The most significant issue with this paper is its failure to properly acknowledge and position itself with respect to existing work that is nearly identical in methodology and motivation. In particular, the ICLR 2023 paper[1] that proposes a hyper label model for programmatic weak supervision that is, in essence, the same as the approach in this paper:
  - Both solve the label aggregation problem.
  - Both aim for cross-dataset generalization and retraining-free inference.
  - Both papers propose a GNN-based model that is pretrained on synthetic data to learn a generalizable label aggregation function.
  - Both use size-invariant initializations and GNN architectures to handle variable numbers of annotators (workers/LFs) and tasks.
  - Both demonstrate that their model, once trained, can be applied to new datasets in a single forward pass, outperforming dataset-specific methods in both accuracy and efficiency.

While the paper claims to be the "first foundation model for label aggregation," this is not accurate. The hyper label model[1] is a direct precedent, and the technical similarities are substantial. The only notable difference is the application domain: The hyper label model[1] focuses on programmatic weak supervision (labeling functions), while this paper focuses on crowdsourcing (human workers). However, the mathematical setup is essentially the same (a bipartite label matrix with noisy annotators) in these two domains.
The lack of discussion, comparison, or even citation of this highly relevant prior work is a serious omission. This gives the misleading impression that the proposed approach is novel, when in fact it is a straightforward adaptation of an existing method to a closely related domain.

2. The paper repeatedly claims to be the "first" to propose a foundation model for label aggregation, and to introduce a new paradigm. Given the existence of[1], these claims are overstated.

3.  Given the existence of the hyper label model approach[1], the main contribution of the paper seems to be modifications to the synthetic data generator and GNN architecture.

[1] Wu et al., "Learning Hyper Label Model for Programmatic Weak Supervision" (ICLR 2023)

**Questions:**

1. Are the authors aware of the hyper label model[1]? Can the authors clarify the technical and conceptual differences between CrowdFM and the hyper label model, beyond the application domain? Please explicitly discuss this prior work in the related work section, and provide a detailed comparison (both conceptual and empirical) to clarify what is novel in CrowdFM.
2. How does CrowdFM perform compared to the hyper label model?
3. What new insights does CrowdFM provide for the crowdsourcing setting on top of what is discussed in[1].
4. Is CrowdFM able to provide any theoretical guarantees as in[1].

[1] Wu et al., "Learning Hyper Label Model for Programmatic Weak Supervision" (ICLR 2023)

---

> ### Author Response · Authors · 2025-11-21
>
> We sincerely thank the reviewer for the thoughtful feedback and for highlighting HyperLM. **We clarify that its omission in the original submission was unintentional.** HyperLM originates from the programmatic weak supervision literature, which is typically considered a separate research direction from human crowdsourcing. As our initial related work focused specifically on crowdsourcing methods, HyperLM was not included.
>
> In response to the reviewer’s comment, we have carefully reviewed the HyperLM paper, conducted additional experiments comparing it with CrowdFM under comparable settings, and further surveyed the broader landscape of related approaches in weak supervision. We apologize for the oversight and have added a comprehensive discussion of HyperLM in the revised manuscript, with comparisons in the Introduction, Experiments, Related Work, and Appendix H.
>
> Below, we organize our response into two parts: conceptual differences in modeling and design, and empirical results from extensive experiments. Together, they demonstrate that CrowdFM is purpose-built as a foundation model for human crowdsourcing, while HyperLM, despite superficial similarities, is not suited for this setting.
>
> ## Conceptual Differences
>
> ### Crowdsourcing foundation model
>
> A true foundation model for crowdsourcing should meet two core criteria: 1) it must scale efficiently to datasets of large size; 2) it must learn transferable representations for downstream use. HyperLM does not satisfy these requirements.
> - **Scalability:** HyperLM constructs the graph based on individual binary annotations. This leads to high computational cost when annotations are dense. For multi-class tasks, it requires building K separate graphs, resulting in overhead that scales with label cardinality. This design limits its ability to generalize to large-scale datasets, which are central to the foundation model paradigm. In contrast, CrowdFM uses a compact graph over workers, tasks and options, enabling efficient inference regardless of label cardinality or annotation density.
> - **Transferable representations:** HyperLM does not explicitly model workers or tasks and Its synthetic data generation relies on uniform label sampling, which fails to capture critical aspects of real-world annotation behavior, such as worker reliability and task difficulty. Consequently, it cannot produce meaningful or reusable behavioral representations. In contrast, CrowdFM leverages synthetic data generation grounded in the Three-Parameter Logistic (3PL) item response theory and employs a GNN to model workers and tasks as explicit nodes in the graph. This enables CrowdFM to capture crowd heterogeneity effectively and learn structured, transferable representations that facilitate seamless adaptation to downstream applications.
> ### Synthetic data generation
>
> The effectiveness of synthetic data in training generalizable models hinges on its distributional fidelity to real-world annotations. To quantify this, we measure distributional similarity using Wasserstein distance across 16 statistical metrics and 22 real-world crowdsourcing datasets.
>
> CrowdFM's generator achieves average Wasserstein distances below 0.15, indicating a high degree of alignment with empirical annotation patterns (Appendix F). In contrast, datasets generated using HyperLM's uniform sampling strategy exhibit distances typically ranging from 0.3 to 0.6 (Appendix H), revealing **a significant divergence from real-world behavior**.
>
> Models trained on such unrealistic synthetic data tend to learn independence or uniform-error assumptions, which limits their effectiveness in real crowdsourcing environments. CrowdFM bridges the sim-to-real gap through behaviorally grounded simulation.

---

> > ### Author Response · Authors · 2025-11-21
> >
> > ## Experimental Results
> >
> > ### Performance on human crowdsourcing datasets
> >
> > To address the reviewer's question directly, we conducted extensive empirical comparisons. We evaluated HyperLM on all 22 crowdsourcing datasets and updated Table 1 and Appendix E accordingly. **HyperLM performs substantially below CrowdFM, with average accuracy 80.81%, lower than MV (81.78%) and significantly lower than CrowdFM (83.41%)**. The Wilcoxon signed-rank test gives p=0.01793, showing that CrowdFM significantly outperforms HyperLM. Runtime differences also reflect scalability limitations: **on the large Senti dataset, HyperLM requires 16.72 seconds, whereas CrowdFM requires only 5.75 seconds**. HyperLM wins only 12 out of the 22 datasets, compared with CrowdFM's 21.
> >
> > | **Method**  | #Win↑ | Acc.↑ | Runtime↓ | P-value |
> > | ----------- | ----- | ----- | -------- | ------- |
> > | **MV**      | -     | 81.78 | 0.04     | 0.00003 |
> > | **PM**      | 13    | 80.27 | 0.47     | 0.00647 |
> > | **CATD**    | 15    | 83.06 | 2.59     | 0.20700 |
> > | **DS**      | 16    | 83.02 | 5.24     | 0.31889 |
> > | **BWA**     | 17    | 83.31 | 0.10     | 0.60871 |
> > | **IBCC**    | 15    | 83.07 | 0.12     | 0.36658 |
> > | **EBCC**    | 17    | 84.08 | 2.95     | 0.90089 |
> > | **GLAD**    | 16    | 82.75 | 494.26   | 0.19475 |
> > | **LAA**     | 10    | 78.42 | 223.06   | 0.04935 |
> > | **TiReMGE** | 6     | 80.29 | 26.77    | 0.00230 |
> > | **GOVERN**  | 13    | 82.61 | 95.43    | 0.28992 |
> > | **HyperLM** | 12    | 80.81 | 0.88     | 0.01793 |
> > | **CrowdFM** | 21    | 83.41 | 0.53     | -       |
> > ### Performance on programmatic weak supervision datasets
> >
> > To ensure fairness, we also evaluated CrowdFM on HyperLM's own programmatic weak supervision benchmarks . **Despite CrowdFM not being tailored for this domain, its average accuracy (75.70%)  exceeds HyperLM (75.15%)**, as shown below. This suggests that the behavioral inductive biases learned by CrowdFM transfer moderately to weak supervision, whereas HyperLM's do not transfer to human annotation.
> >
> > | Method  | Average   | Bioresponse | PhishingWebsites | agnews | bank-marketing | basketball | cdr   | census | chemprot | commercial | imdb  | mushroom | semeval | sms   | spambase | spouse | tennis | trec  | yelp  | youtube |
> > | ------ | -------- | ---------- | --------------- | ----- | ------------- | --------- | ---- | ----- | ------- | --------- | ---- | ------- | ------ | ---- | ------- | ----- | ----- | ---- | ---- | ------ |
> > | **HyperLM** | **75.15** | 59.69       | 78.35            | 81.38  | 67.02          | 62.32      | 72.58 | 79.39  | 52.32    | 88.20      | 88.52 | 87.11    | 84.21   | 93.86 | 73.61    | 60.20  | 87.20  | 60.02 | 74.42 | 90.99   |
> > | **CrowdFM** | **75.70** | 54.53       | 78.71            | 81.55  | 75.43          | 76.89      | 72.46 | 79.81  | 54.05    | 90.92      | 72.96 | 86.97    | 83.42   | 93.94 | 74.80    | 61.39  | 87.01  | 52.31 | 72.78 | 87.94   |

---

> > > ### Comment · Reviewer_QfN2 · 2025-11-21
> > >
> > > Thank you for sharing the additional experiment results. One important detail for evaluation is that some datasets have highly imbalanced binary labels or domain-specific requirements, making accuracy an unreliable measure of performance. In existing benchmarks, it is common practice to use the F1 metric instead. For example, the F1 metric is typically adopted for datasets such as SMS, Spouse, CDR, Commercial, Tennis, and Basketball [1,2]. It would be helpful if you could update the evaluation metric to F1, so that the results are directly comparable to those in previous work.
> > >
> > > [1] Zhang, Jieyu, et al. "WRENCH: A Comprehensive Benchmark for Weak Supervision." Thirty-fifth Conference on Neural Information Processing Systems Datasets and Benchmarks Track (Round 2).
> > > [2] Wu et al., "Learning Hyper Label Model for Programmatic Weak Supervision" (ICLR 2023)

---

> > > > ### Author Response · Authors · 2025-12-04
> > > >
> > > > We thank the reviewer for pointing out the importance of using F1 on weak supervision benchmarks, and we fully agree. Following the established practice in WRENCH and HyperLM, we have re-evaluated  CrowdFM using F1 and accuracy on the appropriate datasets, as summarized in the table below. Across all 14 datasets, CrowdFM achieves an average of 68.4%, closely matching HyperLM's 69.0%. More importantly, CrowdFM outperforms MV on 9 datasets, while HyperLM outperforms MV on 5 and equal to MV on 5. These results confirm that **CrowdFM transfers well to programmatic weak supervision, despite not being designed for this domain, while HyperLM does not transfer effectively to human annotation (as shown in our main crowdsourcing experiments).**
> > > >
> > > > | Dataset | #Win | AVG.     | AGNews | Basketball | CDR  | Census | ChemProt | Commercial | IMDB | SemEval | SMS  | Spouse | Tennis | TREC | Yelp | Youtube |
> > > > | ------- | ---- | -------- | ------ | ---------- | ---- | ------ | -------- | ---------- | ---- | ------- | ---- | ------ | ------ | ---- | ---- | ------- |
> > > > | MV      | -    | 65.0±0.0 | 81.4   | 18.9       | 63.3 | 22.2   | 53.7     | 85.9       | 75.0 | 84.2    | 84.0 | 51.6   | 85.0   | 49.9 | 74.4 | 80.3    |
> > > > | DP      | 4    | 60.6±0.1 | 81.7   | 17.1       | 33.9 | 11.1   | 56.2     | 77.5       | 74.4 | 73.5    | 83.8 | 50.3   | 85.1   | 47.2 | 71.9 | 84.5    |
> > > > | FS      | 3    | 59.6±0.0 | 81.3   | 17.1       | 69.6 | 17.1   | 52.4     | 82.5       | 74.5 | 23.8    | 74.4 | 49.9   | 84.0   | 50.1 | 74.0 | 83.7    |
> > > > | MeTaL   | 6    | 65.5±0.2 | 82.2   | 19.0       | 67.9 | 51.1   | 52.9     | 83.7       | 75.0 | 84.2    | 57.7 | 49.9   | 80.9   | 52.1 | 74.4 | 86.0    |
> > > > | NPLM    | 0    | 40.1±0.0 | 81.3   | 0.0        | 0.0  | 0.0    | 48.4     | 76.5       | 55.2 | 30.2    | 0.0  | 34.3   | 85.0   | 36.5 | 68.3 | 45.2    |
> > > > | DS      | 0    | 44.5±0.0 | 26.6   | 17.1       | 0.1  | 0.0    | 35.1     | 77.8       | 74.4 | 73.5    | 65.0 | 34.3   | 85.0   | 20.9 | 68.3 | 45.2    |
> > > > | EBCC    | 0    | 37.6±0.1 | 27.8   | 17.1       | 8.7  | 0.0    | 35.0     | 77.5       | 74.4 | 30.2    | 0.0  | 34.3   | 85.0   | 20.8 | 69.6 | 45.2    |
> > > > | CLL     | 5    | 67.6±0.0 | 80.7   | 17.5       | 64.9 | 53.6   | 53.1     | 84.8       | 72.7 | 84.2    | 84.2 | 50.0   | 83.5   | 59.0 | 72.0 | 86.1    |
> > > > | HyperLM | 5    | 69.0±0.2 | 81.4   | 17.1       | 71.0 | 56.1   | 52.3     | 83.6       | 75.0 | 84.2    | 84.1 | 51.6   | 84.3   | 59.8 | 74.4 | 91.4    |
> > > > | CrowdFM | 9    | 68.4±0.2 | 81.6   | 23.3       | 66.4 | 28.0   | 54.1     | 84.9       | 73.4 | 83.4    | 96.5 | 69.2   | 84.0   | 52.3 | 72.7 | 87.9    |

---

> > ### Comment · Reviewer_QfN2 · 2025-11-21
> >
> > Thank you to the authors for the detailed response, and for conducting additional experiments and expanding the discussion in the revised manuscript. I want to be clear that I would be happy to recommend acceptance if the paper is properly positioned and prior work is fairly acknowledged. To facilitate a productive revision, I would like to reiterate several key points regarding the conceptual positioning and claims of the work:
> >
> > 1. On the claimed paradigm shift. The central conceptual advance, i.e. moving from dataset-specific parameter estimation to a universal retraining-free aggregation model, was first introduced by HyperLM. Furthermore, the overall technical framework for achieving such paradigm i.e. (1) synthetic dataset generation followed by (2) training graph neural network to achieve invariance to input size, was first introduced by HyperLM and adopted by CrowdFM. While CrowdFM brings important technical improvements and domain-specific adaptations for crowdsourcing (such as more realistic synthetic data generation, explicit modeling of workers and tasks, and a new graph structure), these are technical design choices within the same paradigm. Therefore, CrowdFM should not claim to be the originator of the "universal aggregation paradigm" in label aggregation. The paradigm shift itself was established by HyperLM, and CrowdFM is best positioned as an extension and improvement tailored to the crowdsourcing domain. I strongly encourage the authors to revise the introduction and positioning to accurately reflect this.
> >
> > 2. The scalability concerns raised about HyperLM are primarily implementation details. HyperLM's GNN architecture is capable of supporting multi-class labels, though its synthetic data generation is limited to binary labels. CrowdFM's support for multi-class synthetic data generation is indeed an advance, but this represents an improvement upon the existing paradigm, not a new paradigm. In terms of efficiency, HyperLM requires multiple forward passes for multi-class inference (one per class), which may result in K times the inference time compared to CrowdFM, but memory usage is similar. As shown in the new experiments, the difference in inference time (e.g., 16.72s for HyperLM vs. 5.75s for CrowdFM on a large dataset) is not substantial for label aggregation tasks, which are typically not time-sensitive. Thus, while CrowdFM's design is more efficient for multi-class and dense annotation scenarios, this is a technical improvement rather than a conceptual shift.
> >
> > 3. For an accurate and fair representation of the literature, HyperLM should be acknowledged as the prior foundation model that first implemented the paradigm of "achieving the accuracy of sophisticated models while retaining the scalability and retraining-free nature of Majority Voting" for label aggregation. In addition, HyperLM should be acknowledged as the first work that proposed the overall technical framework (i.e. (1) synthetic dataset generation followed by (2) training graph neural network to achieve invariance to input size) for achieving such paradigm, which is adopted by CrowdFM. CrowdFM can then claim to first achieve this paradigm for crowdsourced label aggregation, with improved performance and domain-specific adaptations. This framing would avoid over-claiming and fairly represent prior work.
> >
> > To summarize, while CrowdFM makes meaningful technical advances for crowdsourcing label aggregation, the conceptual paradigm shift was first realized by HyperLM. I encourage the authors to revise their claims and positioning to reflect this, and to clearly acknowledge HyperLM. This will strengthen the value and integrity of the paper.

---

> > > ### Author Response · Authors · 2025-12-04
> > >
> > > We thank the reviewer for the clear guidance on properly positioning our work. In the revised manuscript, we explicitly acknowledge HyperLM as the prior universal aggregation model that first established this paradigm. We now clarify that our contribution is to advance this paradigm in the crowdsourcing setting, which involves structural heterogeneity and behavioral patterns not addressed by HyperLM.

---

### Official Review · Reviewer_7XU2 · 2025-10-31

**Soundness:** 2
**Presentation:** 2
**Contribution:** 3
**Rating:** 6
**Confidence:** 4

**Summary:**

This paper proposes CrowdFM, the first foundation model for crowdsourced label aggregation. It replaces traditional per-dataset parameter estimation with a pretrained bipartite graph neural network trained on a vast, domain-randomized synthetic dataset. By leveraging a size-invariant initialization and attention-based message passing, it learns universal principles of collective intelligence and generalizes to new, unseen datasets. Experiments on 22 real-world datasets show that CrowdFM matches or outperforms state-of-the-art methods in both accuracy and efficiency, while its learned representations also support worker assessment and task assignment.

**Strengths:**

1.The proposed CrowdFM is the first foundation model for label aggregation, enabling cross-dataset generalization without retraining and marking a key step toward universal label aggregation.

2.CrowdFM further exhibits strong versatility, as its pretrained representations can be directly adapted to multiple downstream tasks such as worker assessment and task assignment without additional training.

3.Extensive experiments on 22 real-world benchmarks demonstrate that the proposed single, fixed model consistently matches or surpasses dataset-specific methods in both accuracy and efficiency.

**Weaknesses:**

1.The paper lacks a systematic quantitative analysis of synthetic and real-world crowdsourced data. Although the authors validate generalization on real datasets, they do not analyze the distributional differences of key parameters (e.g., worker ability θ, task difficulty β, task guessing rate c) between synthetic and real data, making it difficult to assess how well the learned patterns reflect real human annotation behaviors. Besides, how to obtain these sampling ranges?

2.Although the parameters in the synthetic data generator are randomly sampled, the paper lacks a sensitivity analysis of key parameters. Random sampling alone cannot fully verify the model’s robustness across different crowdsourcing conditions. It is recommended to include such an analysis to strengthen the credibility of CrowdFM’s generalization claims.

3.With regard to the comparison results, statistical tests are needed in the comparison results. The detailed description about statistical tests for comparisons of multiple algorithms on multiple datasets can be found from the papers such as Statistical comparisons of classifiers over multiple data sets.

4.The paper lacks ablation studies to verify the individual contributions of key components (e.g., attention-based aggregation, size-invariant initialization, and synthetic data diversity). Without such analysis, it is difficult to determine which design choices are most critical to the model’s performance gains.

**Questions:**

The same as the weaknesses above.

---

> ### Author Response · Authors · 2025-11-21
>
> We sincerely thank the reviewer for the thoughtful evaluation and constructive feedback. Below, we address each of your concerns in detail.
>
> > W1: Lack of systematic quantitative analysis between synthetic and real data
>
> We acknowledge the importance of evaluating the alignment between synthetic and real-world data. Since latent factors such as worker ability and task difficulty are not directly observable, we compare behavioral proxies including accuracy and load ratio across datasets.
>
> **We have added a new analysis in Appendix F to evaluate the fidelity of our synthetic data generator.** We sampled 10,000 synthetic datasets from the domain-randomized generator and collected statistical profiles from 22 real-world datasets. For each dataset, we computed 16 descriptive metrics, including means, standard deviations and percentiles, over key annotation proxies. **The Wasserstein distance between the synthetic and real distributions is consistently below 0.15 across all metrics, indicating strong distributional alignment.** This demonstrates that our generator effectively captures the core statistical characteristics of real-world annotation behaviors.
>
> > W2: Lack of sensitivity analysis on key generator parameters
>
> Thank you for this suggestion. **We have added a new experiment in Section 4.4 to analyze the impact of the GNN's depth (L) and hidden dimension (d).** Our findings show that model performance generally improves as L and d increase, and even with a smaller configuration (e.g., L=6, d=16), CrowdFM achieves performance that is superior to MV.
>
> > W3: Need for statistical significance testing in comparisons
>
> We appreciate the feedback and have incorporated statistical significance testing to strengthen our comparisons. **We have added a statistical significance analysis to Table 1, reporting p-values from the Wilcoxon signed-ranks test for each method relative to CrowdFM.** As reported, CrowdFM achieves significantly higher performance than MV, PM, LAA, TiReMGE, and HyperLM, confirming the effectiveness of our approach. While EBCC shows marginally higher average accuracy, the difference is not statistically significant (p=0.90089 ).
>
> > W4: Lack of ablation studies on key components
>
> We agree that understanding component contributions is important. **We have added an ablation study in Section 4.4**, evaluating the core design choices: the domain-randomized synthetic data generator and the GNN's attention mechanism. Results show that removing either leads to a significant performance drop. Replacing the generator with a simpler uniform approach reduces accuracy, demonstrating the importance of diverse synthetic data for sim-to-real transfer. Removing attention degrades performance, confirming its role in modeling annotation heterogeneity.

---

> > ### Comment · Reviewer_7XU2 · 2025-11-25
> >
> > Thanks for the authors' rebuttal and revised paper. I have decided to raise my score to 8.

---

### Official Review · Reviewer_BaKD · 2025-11-01

**Soundness:** 3
**Presentation:** 2
**Contribution:** 2
**Rating:** 4
**Confidence:** 3

**Summary:**

Traditional methods (e.g., Dawid–Skene, GLAD, EBCC) typically adopt a dataset-specific modeling paradigm: they require training model parameters from scratch for each new dataset, resulting in poor generalization. Meanwhile, Majority Voting (MV), widely used in industry, requires no training but ignores differences in annotators' capabilities, leading to limited accuracy.
To bridge this gap, the authors propose a universal aggregation paradigm: by pre-training a bipartite GNN on large-scale, domain-randomized synthetic datasets, the model learns transferable "collective intelligence" aggregation rules. During the inference phase, the model can be directly applied to any real crowdsourced dataset without fine-tuning or retraining.

**Strengths:**

1.	Strong paradigm innovation: For the first time, the foundation model concept was introduced into crowdsourced label aggregation, breaking the dataset-specific paradigm that has persisted for decades.
2.	The problem definition has practical significance: Shifting the aggregation of crowdsourced labels from the "dataset-by-dataset modeling" paradigm to the "unified fundamental model" paradigm aligns with the urgent demand of the industrial sector for scalable and retraining-free systems.
3.	The experiments were thorough: 22 real datasets covered text, images, and audio, and included extreme scale/density scenarios, verifying the generalization ability.
4.	The accuracy rate and running time were reported, and it was pointed out that some methods failed due to insufficient memory

**Weaknesses:**

1.	How can we ensure that the distribution of synthetic data is sufficient to cover the feature space of real crowdsourcing tasks? Is there any performance degradation under certain types of tasks (such as extremely unbalanced ones)?
2.	Can CrowdFM be regarded as the specialization of GraphFM in the field of crowdsourcing? Has the pre-training objective of the existing GFM been borrowed?
3.	The performance for extremely large-scale data (such as Senti and Fact) slightly decreases, but the reasons are not discussed.
4.	The absence of ablation experiments makes the contributions of each module unclear
5.	Although it includes recent works such as EBCC, GOVERN, TiReMGE, etc., some new methods for 2024-2025 are missing (such as Zhang et al., KFNN and IWBVT of NeurIPS 2024 are cited but not used as baselines);
6.	The superparameters such as the number of layers L and dimension d of GNN were not discussed in detail.
7.	The "Sim-to-Real Gap" between synthetic data and real data has not been fully quantified
8.	Has the model truly learned the "principle of collective intelligence", or has it merely fitted the statistical patterns of synthetic data? How to prove.

**Questions:**

Same as weaknesses

---

> ### Author Response · Authors · 2025-11-21
>
> We sincerely thank the reviewer for their thoughtful and constructive feedback. Below, we address each of your concerns point-by-point.
>
> > W1: Coverage of synthetic data and performance on diverse cases & W7: Quantifying the sim-to-real gap
>
> We thank the reviewer for raising this critical point about the sim-to-real gap. To rigorously address this concern, **we have conducted a comprehensive quantitative analysis, the results of which are now presented in Appendix F**.
>
> Specifically, since real-world datasets do not provide the latent parameters (e.g., worker ability, task difficulty) used in our generator, we cannot compare them directly. Instead, we use observable proxies: accuracy and load ratio. We sampled 10,000 synthetic datasets from our domain-randomized generator and computed 16 key metrics for both workers and tasks (mean, std, Q25, Q75 of accuracy/load ratio). We then calculated the Wasserstein distance between the distributions of these metrics in the synthetic data and the 22 real-world benchmark datasets. **The results show that all Wasserstein distances are below 0.15, indicating a strong statistical alignment between our synthetic data and real-world scenarios.** This suggests our generator effectively covers the feature space of typical crowdsourcing tasks.
>
> Regarding performance under diverse conditions (e.g., highly imbalanced labels), we generated 1,000 additional synthetic datasets where we systematically varied five key properties: worker ability, task difficulty, answer density, category distribution entropy, and number of options. **As shown in Appendix G, CrowdFM consistently outperforms MV on most datasets cross all configurations, demonstrating its robustness and effectiveness even under challenging and diverse conditions.**
>
> > W2: Relation to GraphFM
>
> We emphasize that CrowdFM is not a specialized variant of GraphFM, despite both approaches utilizing graph neural networks. They are designed to tackle fundamentally distinct problems, featuring unique architectures and objectives.
>
> GraphFM focuses on developing a universal model capable of generalizing across diverse graph domains such as molecular structures and social networks that exhibit heterogeneous node features and topological structures. Its primary challenge lies in cross-domain generalization under varying input formats and feature richness.
>
> In contrast, CrowdFM targets a specific application: aggregating labels in crowdsourcing settings. It provides a unified framework that adapts to scenarios of any scale including variable numbers of tasks, workers and answer options and accommodates diverse annotation patterns such as differing annotation densities, worker qualities ad task difficulty. The input to CrowdFM is always a bipartite graph with exactly two node types: workers and tasks. Crucially, unlike GraphFM, which relies on rich, pre-defined node features, CrowdFM starts with minimal information only node IDs and learns meaningful embeddings entirely from observed annotation behavior through a process of synthetic pretraining.
>
> > W3: Performance on Senti
>
> The slight performance degradation observed on Senti can be attributed to the domain shift. **As shown in Appendix F, the Senti datasets exhibit statistical profiles that are unusual both compared to our synthetic training data and to most real-world datasets.** Despite this discrepancy, CrowdFM still performs competitively, matching or exceeding the performance of MV. In contrast, most other sophisticated dataset-specific methods fall significantly behind MV on Senti. This underscores our model's robustness and generalization capability.
>
> > W4: Ablation study
>
> We agree that ablation studies are crucial. In response, **we have added an ablation study in Section 4.4.**
>
> Our experiments analyze the impact of two key components: (1) the synthetic data generation strategy (comparing our domain-randomized approach to a simpler, uniform method) and (2) the attention mechanism in the GNN (comparing it to simpler aggregation functions like mean pooling).
>
> The results show that removing either component leads to a significant drop in performance.
> Specifically, using a non-domain-randomized generator significantly reduces accuracy, highlighting that bridging the sim-to-real gap through diverse synthetic data is essential. Similarly, replacing the attention mechanism degrades performance, validating its role in effectively modeling annotation heterogeneity.

---

> ### Author Response · Authors · 2025-11-21
>
> > W5: Baselines of KFNN, IWBVT
>
> We appreciate the suggestion to include these recent methods. However, a direct comparison is not feasible due to fundamental differences in their requirements. **IWBVT requires a set of ground-truth labels for calibration, and KFNN relies on side information such as node representations derived from raw inputs.** In contrast, our method operates in a fully unsupervised manner, using only the annotation matrix without any auxiliary data or labeled instances. This aligns with the standard aggregation framework used by classical and state-of-the-art methods such as DS, EBCC, and GOVERN.
>
> > W6: Discussion of GNN hyperparameters (L, d)
>
> Thank you for this suggestion. **We have added a new experiment in Section 4.4 to analyze the impact of the GNN's depth (L) and hidden dimension (d).** Our findings show that model performance generally improves as L and d increase, and even with a smaller configuration (e.g., L=6, d=16), CrowdFM achieves performance that is superior to MV.
>
> > W8: Has the model really learned "collective intelligence principles"?
>
> Our experiments show that CrowdFM matches or outperforms MV and many dataset-specific methods across 22 diverse real-world datasets. As shown in Appendix G, it maintains strong performance under varying conditions, including differences in worker reliability, task difficulty, annotation density, and label imbalance, demonstrating robust generalization.
>
> Moreover, CrowdFM's learned representations are meaningful and transferable. They effectively capture heterogeneity in worker and task behavior, support downstream tasks such as worker evaluation and task assignment. By understanding patterns like reliable worker behavior and how task difficulty affects consistency, CrowdFM enables robust aggregation by inferring latent patterns from observed annotation behavior.

---

### Official Review · Reviewer_mWdt · 2025-11-05

**Soundness:** 3
**Presentation:** 3
**Contribution:** 3
**Rating:** 8
**Confidence:** 3

**Summary:**

The paper aims at building a foundation model for crowdsourced label aggregation that generalizes across heterogeneous datasets without dataset-specific training. It first generates a synthetic crowdsourced data generator to produce diverse synthetic datasets, and then trains a bipartite graph neural network which can be generalizable to other data and tasks. Experiments show the foundation model is efficient and achieve performance comparable to or superior to state-of-the-art aggregation methods.

**Strengths:**

1. Generating the synthetic data and then train a generalizable GNN model to approximate it is a smart idea to unify the data-specific crowdsourcing models. I think it is essentially a type of knowledge distillation, which distills the knowledge from the data generator to a GNN with less input features (e.g. task difficulty/worker ability, which are generally regarded as latent parameters in traditional probabilistic models).
2. The method is sound, and the presentation is clear.
3. Experiments show good accuracy with more efficiency.

**Weaknesses:**

1. The backbone model is relatively simple and not new, but the design is a good fit for crowdsourcing, e.g. the size-invariant initialization for worker/task emebddings. I do not see it as a weakness, but just not a novel contribution. I still like the whole framework of building synthetic data and train a simple domain-indepedent model as foundation models.

2. The possible limitation of the model is it may lack the flexibility to adapt to the case when workers/tasks have their attributes, since these nodes are initialized with fixed pretrained embeddings.

3. Another possible weakness is the limitation of the data generator. Although it is a quite general scheme, there might be other graphical models that it cannot cover; and also the prior distribution of the parameters might cause some bias for the pretrained foundation model.

**Questions:**

See weaknesses.

---

> ### Author Response · Authors · 2025-11-21
>
> We sincerely thank the reviewer for the constructive and encouraging feedback. Below we address the raised concerns point by point.
>
> > W1: Backbone model is relatively simple
>
> We agree that the backbone architecture is intentionally simple. Our contribution does not lie in introducing a brand new GNN variant, but in demonstrating that a minimal, size-invariant architecture combined with domain-randomized synthetic pretraining can serve as a foundation model for crowdsourcing.
>
> > W2: Adaptation to the case when workers/tasks have their attributes
>
> Our current design uses size-invariant initialization because most crowdsourcing datasets do not provide worker or task attributes. We acknowledge that worker and task representations may carry diverse and domain-specific structural meanings, which remains a challenge in general graph learning. A promising direction is to introduce a embeddings alignment module that maps heterogeneous attributes into a unified embedding space.
>
> > W3: Limitations of the data generator
>
> We agree that no single generator can cover all possible graphical models and that synthetic priors may introduce bias. Addressing the sim-to-real gap is indeed an important area for future improvement.
> To evaluate our current generator, **we conducted an additional sim-to-real analysis as reported in the updated Appendix F**. We sampled 10,000 synthetic datasets and compared them with the 22 real datasets across 16 metrics for both workers and tasks (mean, std, Q25, Q75 of accuracy/load ratio). The results show that the synthetic distributions largely cover and resemble those in the real data, with Wasserstein distances consistently below 0.15. This indicates that although there is room for further enhancement, **the current generator already captures most real-world patterns and supports robust cross-dataset generalization.**

---

### Author Response · Authors · 2025-12-04

Dear AC, Dear Reviewers,

We sincerely thank the AC for handling our submission and all reviewers for their constructive feedback. We have thoroughly revised the paper and highlighted all changes in blue.
## Summary of Our Work
This paper introduces **CrowdFM, a foundation model specifically designed for crowdsourced label aggregation**. Unlike traditional methods that must retrain per dataset, CrowdFM is:
- **Universal**: a single pre-trained GNN that generalizes efficiently to any configuration crowdsourcing dataset without retraining.
- **Sim-to-real consistent**: pre-trained on a domain-randomized synthetic corpus that statistically resembles real annotation behavior, including worker reliability, task difficulty, and sparsity.
- **Representation-transferable**: the learned representations support multiple downstream adaptation such as worker evaluation and task assignment.
Across 22 real-world datasets, CrowdFM matches or surpasses state-of-the-art dataset-specific methods while maintaining fast inference. In addition, we introduced comprehensive sim-to-real quantification, ablation studies, and statistical significance tests.
## Response to Key Concerns in the Rebuttal
During the rebuttal, reviewers raised several important questions that helped us substantially strengthen the paper. Below, we summarize the key concerns and acknowledge the reviewers who brought them up.
1. `Coverage of the synthetic data generator` (Reviewer mWdt, BaKD, 7XU2): We added a thorough sim-to-real analysis using Wasserstein distances computed over 16 behavioral statistics across 22 real datasets. As shown in **Appendix F,** all distances remain below 0.15, demonstrating strong distributional alignment between synthetic and real annotation behaviors.
2. `Ablation studies and hyperparameter sensitivity` (Reviewer BaKD, 7XU2): We added new ablation experiments and hyperparameter analyses in **Section 4.4**, showing (1) clear performance drops when removing key components, and (2) stable performance gains as model depth and width increase. These analyses clarify the contribution of each design choice.
3. `Discussion of HyperLM` (Reviewer QfN2): We added a full conceptual and empirical comparison with HyperLM in the Introduction, Related Work, Experiments, and Appendix H. Reviewer QfN2 noted that this comparison is essential for properly positioning our contribution, stating: "I would be happy to recommend acceptance if the paper is properly positioned and prior work is fairly acknowledged." We have now incorporated all the requested analyses exactly as suggested.
## Summary
During the rebuttal, prior to the unexpected incident:
- Reviewer QfN2, despite giving a low initial score, made an extremely helpful observation regarding related work. Through detailed discussion, the reviewer acknowledged the contribution of CrowdFM and indicated that **acceptance would be recommended** once the comparison with HyperLM was fully integrated. We have now fully addressed this feedback in the revised manuscript.
- Reviewer 7XU2 **increased their score to 8** after checking the  rebuttal and revised paper.

We respectfully ask you to review our submission with these updates and reviewer responses in mind. We believe the manuscript is now clearer, more complete, and addresses all major concerns.

Thank you for your time and consideration.

Best regards,

Authors

---

### Meta-Review · Area_Chair_1wfe · 2025-12-26

**Summary:**

Reviewers were generally positive about soundness and the goal of a retraining-free, cross-dataset aggregation model, and the author–reviewer discussion indicates the revision added several requested analyses (sim-to-real quantification, ablations, and statistical testing). My remaining hesitation is primarily about the magnitude and clarity of the contribution: the discussion repeatedly notes that the overall paradigm and framework are closely related to prior amortized aggregation work (especially HyperLM), so the novelty appears more like a crowdsourcing-specific refinement than a new conceptual shift. In addition, while CrowdFM tends to outperform majority voting, the average gain appears small, which makes it harder to judge the practical significance relative to the added complexity. Finally, despite the added analyses, there remains some uncertainty about how broadly the synthetic generator’s coverage supports real-world generalization and about metric choices in imbalanced settings. Overall, this supports a poster recommendation with low confidence—solid execution and improved rigor, but limited perceived conceptual advance and residual uncertainty about generality and impact.

**Reviewer Concerns:**

Concerns addressed in the discussion include the need for stronger evidence that the synthetic generator matches real crowdsourcing behavior (the authors added sim-to-real quantification), missing ablations and hyperparameter/sensitivity analysis (the authors added both), and lack of statistical rigor in the multi-dataset comparisons (the authors added significance tests). The authors also corrected the major omission around HyperLM by adding explicit positioning and head-to-head comparisons, which reviewer QfN2 indicated was the key condition for recommending acceptance.

Concerns still outstanding or only partially resolved are that the work may still read as incremental depending on how convincingly it is framed as an extension of the HyperLM paradigm rather than a new shift, and that real-world applicability remains uncertain in settings where domain/task features or worker/task attributes matter (not directly evaluated). Baseline coverage remains debatable because some suggested recent methods were excluded as “not comparable,” and metric choices under imbalance (accuracy vs F1) were raised but do not appear fully settled across the crowdsourcing results, along with limited “when/why it fails” analysis for atypical or very large datasets

**Reviewer Scores:**

- mWdt (8 → 8): Their concerns were mainly about limited novelty and possible limits when worker/task attributes exist, but the discussion doesn’t suggest a reason to downgrade, and the additional analyses likely reinforce their positive view.

 - BaKD (4 → 5):  The authors directly addressed the requests for sim-to-real quantification, ablations, and hyperparameter/statistical analysis, but baseline coverage/comparability and the strength of the “learned principle” claim still leave it borderline.

 - 7XU2 (6 → 8): They explicitly raised their score to 8 after reading the rebuttal and revised paper.

 - QfN2 (2 → 5 or 6): The authors’ acknowledgement and comparison to HyperLM resolves the main omission, but QfN2’s emphasis that the paradigm originates from HyperLM (and concerns about metric choice under imbalance) would likely keep the recommendation at borderline rather than clearly positive.

---

### Decision · Program_Chairs · 2026-01-26

Accept (Poster)